# ParallelTime: Dynamically Weighting the Balance of Short- and Long-Term Temporal Dependencies

## Abstract

Modern multivariate time series forecasting primarily relies on two architectures: the Transformer with attention mechanism and Mamba. In natural language processing, an approach has been used that combines local window attention for capturing short-term dependencies and Mamba for capturing long-term dependencies, with their outputs averaged to assign equal weight to both. We find that for time-series forecasting tasks, assigning equal weight to long-term and short-term dependencies is not optimal. To mitigate this, we propose a dynamic weighting mechanism, ParallelTime Weighter, which calculates interdependent weights for long-term and short-term dependencies for each token based on the input and the model's knowledge. Furthermore, we introduce the ParallelTime architecture, which incorporates the ParallelTime Weighter mechanism to deliver state-of-the-art performance across diverse benchmarks. Our architecture demonstrates robustness, achieves lower FLOPs, requires fewer parameters, scales effectively to longer prediction horizons, and significantly outperforms existing methods. These advances highlight a promising path for future developments of parallel Attention-Mamba in time series forecasting. The implementation is readily available at: GitHub.

## 1 Introduction

Forecasting is one of the most important tasks in time series analysis. To address this challenge, various architectures have been proposed. The Transformer architecture (Vaswani et al., 2017), which has achieved remarkable success in natural language processing (Brown et al., 2020) and computer vision (Dosovitskiy et al., 2021), has also shown promise in time series forecasting (Nie et al., 2023). Another successful architecture introduced in recent years is the State Space Model (SSM) (Gu et al., 2022; Smith et al., 2023). SSM-based models, such as Mamba (Gu and Dao, 2023), have demonstrated strong performance in time series forecasting (Wang et al., 2024) and other domains.

Each approach has its distinct advantages. Mamba, through its parameter initialization, produces a summary of long-term dependencies (Gu et al., 2020). The latter allows for extraction of the leading features for forecasting, while filtering out the noise in the time series. Attention models, such as the transformer, are highly accurate and excel at capturing complex patterns and interactions across the sequence, enabling robust forecasting performance (Nie et al., 2023). Moreover, in cases of channel independence, where each variable in a multivariate time series is processed separately using the same model weights, attention models demonstrate superior performance on datasets with similar variates series (Nie et al., 2023). In contrast, Mamba models, such as those proposed in Wang et al. (2024), achieve better results on datasets with heterogeneous variates series.

In this paper, we propose a novel method that combines the strengths of Mamba and the attention mechanism by computing both Mamba, which captures long-term dependencies, and a small local window attention, which focuses on short-term dependencies. Recent papers in natural language processing (Dong et al., 2024) tackle this problem by computing

the mean of the values and assigning equal weight to both components. In contrast, our approach weights each component of each token separately. In cases where more long-range dependencies are needed for the prediction, the ParallelTime Weighter gives more weight to the Mamba component. When more short-term dependency predictions are required, more weight is given to the window attention component. Additionally, we leverage registers as domain-specific global context, providing a persistent reference that captures information beyond the input series. We demonstrate that our method is robust and significantly outperforms existing approaches, on almost every benchmark dataset.

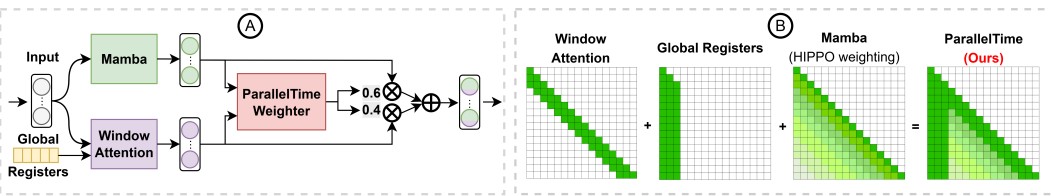

Figure 1: Ⓐ High level visualization of ParallelTime module. Ⓑ Diagram of the attention map of ParallelTime, integrating global registers, local window attention, and Mamba components.

**Our contributions.**  The main contributions of this paper are three-fold:

- We propose a novel ParallelTime Weighter that selects the contributions of short-term, long-term, and global memory for each time series patch, implemented via window-based attention, Mamba, and registers, respectively, to improve the accuracy of long-term forecasting.
- We demonstrate that the parallel Mamba-Attention architecture is the most effective approach for long-term time series forecasting.
- Our model, ParallelTime, achieves SOTA performance on real-world benchmarks, delivering better results from previous models with fewer parameters and lower computational cost, a characteristic highly critical for real-time forecasting applications.

## 2 RELATED WORK

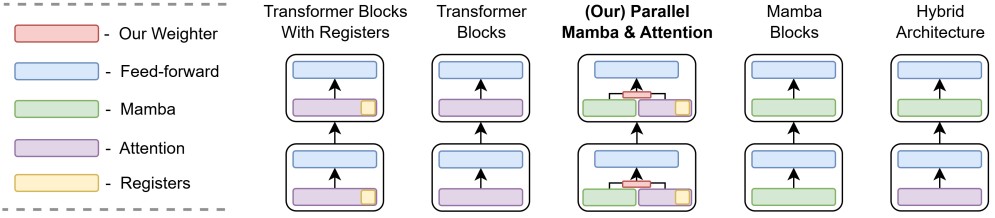

Figure 2: Comparison of five neural network block architectures: Transformer Blocks with Registers, Transformer Blocks, (Ours) Parallel Mamba-Attention with dynamic weighting mechanism , Mamba Blocks, and Hybrid Sequential Mamba-Attention Architecture.

**Transformer** Vaswani et al. (2017), leveraging causal self-attention layers and feed-forward networks, has laid a powerful foundation for time series forecasting (Zhou et al., 2021; 2022). A standout example is PatchTST (Nie et al., 2023), which achieves SOTA performance by utilizing channel independence to process each variable in a multivariate time series separately. By feeding contiguous time series patches as tokens into a standard self-attention mechanism, PatchTST outperforms many previous models. In standard self-attention, each token attends to all preceding tokens to capture global dependencies. To focus on local patterns, windowed attention variants, such as those in LongFormer (Beltagy et al., 2020a) and Swin Transformer (Liu et al., 2021), restrict each token to attend only to the most recent $S$ tokens, as illustrated in Figure 1 (B).

**Registers** are model parameters that function as tokens, concatenated with input tokens to provide additional global domain-specific information. They serve as a persistent reference for the model, capturing information not explicitly present in the input tokens. Registers have shown considerable promise in natural language processing, as demonstrated by Burtsev et al. (2020), and in computer vision, as explored by Darcet et al. (2024), where they enhance model performance by leveraging task-specific memory.

**Mamba** (Gu and Dao, 2023) is a State Space Model (SSM) (Gu et al., 2022; Smith et al., 2023) designed for efficient (Waleffe et al., 2024) and high-performance sequence modeling. At the core of the Mamba architecture is the HIPPO matrix (Gu et al., 2020), which prioritizes recent tokens by assigning them greater influence in the state representation while compressing older tokens into a compact, approximated summary. This approach effectively captures a condensed representation of long-range dependencies, making it well-suited for time series forecasting. S-Mamba (Wang et al., 2024) has demonstrated competitive performance across several time series forecasting benchmarks.

**Hybrid models** which combine Mamba and Attention layers in a sequential stack, have gained prominence in natural language processing, as demonstrated by models such as Jamba (Team et al., 2024) and Samba (Ren et al., 2024). In time-series forecasting, Heracles (Patro et al., 2024) showcases the versatility and effectiveness of this approach. However, sequential stacking may introduce information bottlenecks (Dong et al., 2024) and poses challenges in determining the optimal placement of each component, potentially compromising forecasting accuracy.

**Parallel architectures** where Mamba and attention mechanisms process the same input simultaneously and their outputs are combined in some way, have recently been proposed in natural language processing. For instance, Hymba (Dong et al., 2024) proposed aggregating Mamba and attention outputs via simple averaging. However, in time series forecasting, where window attention mechanisms capture short-term dependencies at each layer, and Mamba is responsible for summarizing long-term dependencies, assigning equal weights to both long-term and short-term dependencies may not optimally capture the right amount of each dependency needed for each prediction, especially when time series variates differ significantly. To the best of our knowledge, no prior work has applied parallel Mamba-Attention models to long-term time series forecasting. We demonstrate that our novel weighted aggregation approach, ParallelTime Weighter, outperforms naive combinations, leveraging the strengths of both components to achieve state-of-the-art performance.

## 3 PARALLELTIME

**Problem definition.** In multivariate long-term time series forecasting, the task is to predict future values of multiple interdependent variables based on historical data. Given a multivariate time series $\mathbf{X} = (\mathbf{x}_1, \ldots, \mathbf{x}_T) \in \mathbb{R}^{N \times T}$, where $N$ is the number of variables and $T$ is the number of timestamps, the goal is to forecast $H$ future values $\mathbf{Y} = (\mathbf{x}_{T+1}, \ldots, \mathbf{x}_{T+H}) \in \mathbb{R}^{N \times H}$. Each $\mathbf{x}_t \in \mathbb{R}^N$ represents the observations of $N$ variables at time $t$.

### 3.1 OVERALL ARCHITECTURE

The ParallelTime architecture is illustrated in Figure 3. Our model begins by decomposing the multivariate time series input into $N$ univariate series, leveraging the channel independence framework (Nie et al., 2023). This approach enables all model weights to learn more than one variant, enhancing robustness during testing. To address distribution shifts across different time series, we apply instance normalization (ReVIn) (Kim et al., 2022) to the input. Subsequently, a patching mechanism divides each univariate series into non-overlapping patches, treating each patch as a "token" with features derived from the univariate time series values. We tried overlapping patches, but they increased computational cost without improving accuracy, so they were not used.

To effectively extract both global trend and local trends from each patch, we employ a dual embedding strategy. A linear layer aggregates global information by mixing all data points within the patch, while a Conv1D layer (O'Shea and Nash, 2015) captures local trends within

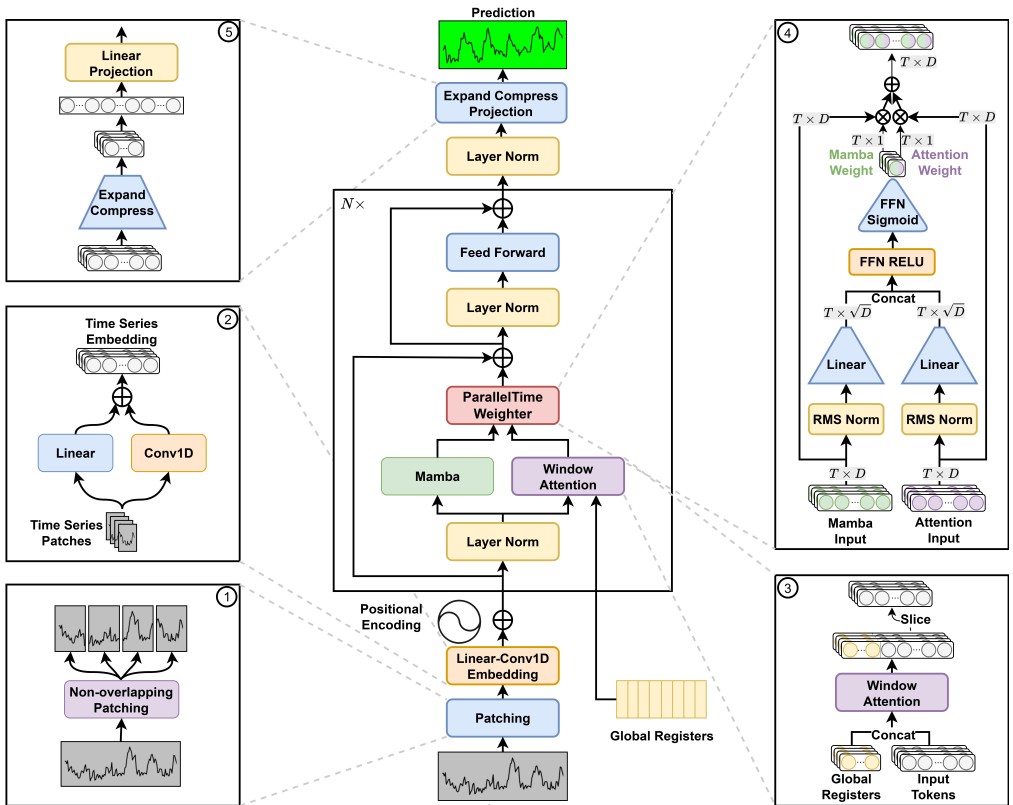

Figure 3: The architecture of ParallelTime. The input time series ① is sliced into non-overlapping patches. ② Each patch is embedded and augmented with positional encoding. The resulting tokens are processed through $N$ stacked ParallelTime blocks. Each block first normalizes the input, applies Mamba computation and ③ windowed attention with a register in parallel, ④ weights their outputs using a ParallelTime Weighter mechanism, and then applies normalization followed by a nonlinear feedforward layer. Finally, the output is normalized ⑤ and processed through an expand-compress-projection mechanism to generate the horizon prediction.

the patch. The global and local representations are then combined through summation to form the final $\mathbf{x}_e$ patch embedding.

To capture sequential order in attention models, which function as a bag of words without positional encoding (Vaswani et al., 2017), we incorporate absolute positional encoding, defined as $\mathbf{x}_d = \mathbf{x}_e + \mathbf{x}_{pos} \in \mathbb{R}^{P \times dim}$, where $\mathbf{x}_e$ represents the input embedding, $\mathbf{x}_{pos}$ denotes the positional encoding, and $\mathbf{x}_d$ is the resulting encoded representation.

## 3.2 PARALLELTIME DECODER BLOCK

Our approach builds upon a decoder-only transformer architecture (Vaswani et al., 2017) As illustrated in Figure 3, the decoder is composed of a stack of $N$ identical layers, with each layer comprising two sublayers. The first sublayer integrates parallel Mamba and attention mechanisms, with their outputs processed by the ParallelTime Weighter, which dynamically allocates weights to the Mamba and attention outputs for each patch or token. The second sublayer is a non-linear feed-forward network with SiLU activation. Each sublayer begins with LayerNorm (Ba et al., 2016) and is enclosed by residual connections (He et al., 2016).

## 3.3 MAMBA AND WINDOWED ATTENTION WITH GLOBAL REGISTERS

**Mamba mechanism.** To achieve high accuracy with low memory requirements, we added a Mamba block (Gu and Dao, 2023), which leverages a state-space model. Mamba's strength lies in its high accuracy (Wang et al., 2024) and constant memory usage, making it ideal for long time series forecasting. The core operation of Mamba is defined as:

$$\mathbf{h}_t = \mathbf{A}\mathbf{h}_{t-1} + \mathbf{B}\mathbf{x}_t, \quad \mathbf{y}_t = \mathbf{C}\mathbf{h}_t,$$

where $\mathbf{x}_t \in \mathbb{R}^{dim}$ is the input at time $t$, $\mathbf{h}_t \in \mathbb{R}^{dim}$ is the hidden state, $\mathbf{y}_t \in \mathbb{R}^{\dim}$ is the output at time $t$, and $\mathbf{A}, \mathbf{B}, \mathbf{C}$ are learnable parameters of the state-space model. The output:

$$\mathbf{x}_{mamba} = \text{Mamba}(\mathbf{x}_d),$$

effectively captures long-range dependencies in the input sequence $\mathbf{x}_d \in \mathbb{R}^{P \times \dim}$.

**Windowed Attention Mechanism.** To capture local interactions efficiently within each layer, we utilize a causal multi-head windowed self-attention mechanism Beltagy et al. (2020b). This approach allows us to restrict attention to a fixed window. We select a small window size, set at a $1:9$ ratio relative to the number of input sequence patches, ensuring that the attention mechanism focuses solely on short-term dependencies while delegating long-term dependencies to Mamba.

**Global Registers.** To incorporate global context, we introduce global register tokens, denoted as $\mathbf{W}_{\text{reg}} \in \mathbb{R}^{R \times dim}$, where $R$ is the number of registers and $dim$ is the embedding dimension. These tokens serve as a compact repository of domain-specific global information, providing the model with access to broader contextual cues. The input sequence $\mathbf{x}_d \in \mathbb{R}^{P \times dim}$, is concatenated with the global registers to form: $\mathbf{x}_{\text{cat}} = \text{Concat}(\mathbf{W}_{\text{reg}}, \mathbf{x}_d) \in \mathbb{R}^{(R+P) \times dim}$. This concatenated sequence is then processed by the causal multi-head windowed attention mechanism, yielding:

$$\mathbf{x}_{att} = \text{WinAtt}(\mathbf{x}_{\text{cat}}).$$

## 3.4 PARALLELTIME WEIGHTER

Considering the Mamba $\mathbf{x}_{mamba}$, which encapsulates both short-term and long-term dependencies, and the window attention $\mathbf{x}_{att}$, which reflects short-term dependencies alongside global dependencies obtained from the registers, to make accurate prediction for different inputs, some inputs need to have more long-term dependency and some need more global, and short-term dependencies, giving a weight to each representation is not enough, we want the weights to be in respect to each other, so we created the novel ParallelTime Weighter.

The attention and Mamba outputs, $\mathbf{x}_{\text{att}}$ and $\mathbf{x}_{\text{mamba}}$, are first normalized using RMSNorm (Zhang and Sennrich, 2019) to address their differing scales. Each output is then processed by a dedicated linear transformation that compresses the dimensionality from dim to $\sqrt{\dim}$ (a choice based on intuition, not cross-validated), preserving essential features:

$$\mathbf{x}'_{\text{att}} = \text{RMSNorm}(\mathbf{x}_{\text{att}})\mathbf{W}_{\text{att}} \in \mathbb{R}^{P \times \sqrt{\dim}},$$

$$\mathbf{x}'_{\text{mamba}} = \text{RMSNorm}(\mathbf{x}_{\text{mamba}})\mathbf{W}_{\text{mamba}} \in \mathbb{R}^{P \times \sqrt{\dim}}.$$

These specialized linear layers effectively tailor the compression to the unique characteristics of the attention and Mamba outputs. The compressed representations are then concatenated to form a unified feature set:

$$\mathbf{x}'_{\text{cat}} = \text{Concat}(\mathbf{x}'_{\text{att}}, \mathbf{x}'_{\text{mamba}}) \in \mathbb{R}^{P \times 2\sqrt{\dim}},$$

Following concatenation, the compressed features from the attention and Mamba branches are processed through a two-layer transformation to capture complex interactions. Inspired by the kernel trick (Hearst et al., 1998), this approach leverages higher-dimensional spaces to reveal patterns not readily discernible in lower dimensions, this step generates adaptive weights:

$$\mathbf{x}_{\text{weights}} = \sigma(\text{ReLU}(\mathbf{x}'_{\text{cat}}\mathbf{W}_1)\mathbf{W}_2) \in \mathbb{R}^{P \times 2}, \mathbf{W}_1 \in \mathbb{R}^{2\sqrt{\dim} \times \text{dim-h}}, \mathbf{W}_2 \in \mathbb{R}^{\text{dim-h} \times 2}$$

where dim-h denotes a dimension higher than $2\sqrt{\text{dim}}$, and $\sigma$ represents the sigmoid function. We attempted to replace the sigmoid function with softmax, but it yielded suboptimal results. This observation aligns with other weight mechanisms, such as the Squeeze-and-Excitation approach (Hu et al., 2018), which also performed better with sigmoid. The weight vector is defined as $\mathbf{x}_{\text{weights}} = [\mathbf{x}_{\text{weight}}^{\text{att}}, \mathbf{x}_{\text{weight}}^{\text{mamba}}]$. The final output is computed as a weighted sum of the original attention and Mamba outputs:

$$\mathbf{x}_{\text{out}} = \mathbf{x}_{\text{att}} \cdot \mathbf{x}_{\text{weight}}^{\text{att}} + \mathbf{x}_{\text{mamba}} \cdot \mathbf{x}_{\text{weight}}^{\text{mamba}},$$

This architecture enables the weights to dynamically balance the contributions of each branch, leading to superior performance, as demonstrated in Table 1.

Following the decoder layers, we apply Layer Normalization (LayerNorm). Unlike standard time series forecasting architectures that simply flatten and project data, our Expand-Compress-Project approach is more efficient. We first expand the data to a higher dimension than the input (dim × higher-dim) and then compress it to a significantly smaller dimension (dim ÷ some-dim) than the input dimension. This approach reduces millions of parameters while maintaining comparable performance (see Appendix 5 for details). The projection output forms our model's prediction, which we then de-normalize using ReVIn (Kim et al., 2022).

## 4 EVALUATIONS

### 4.1 BASELINES AND EXPERIMENTAL SETUP

To assess the performance of our proposed ParallelTime, we compare it against several SOTA models for long time series forecasting. These include Transformer-based models such as PatchTST (Nie et al., 2023) iTransformer (Liu et al., 2024) and FEDFormer (Zhou et al., 2022), Mamba models S-Mamba (Wang et al., 2024), Linear model DLinear (Zeng et al., 2023), foundational models including Moment (Goswami et al., 2024), GPT4TS (Zhou et al., 2023), and TimesNet (Wu et al., 2023). We evaluate all models on eight widely used datasets: Electricity, Weather, Illness, Traffic, and four ETT datasets (ETTh1, ETTh2, ETTm1, ETTm2). For detailed dataset descriptions, see Appendix 8.1.

We adopt standard evaluation protocols with prediction horizons of $T \in \{24, 36, 48, 60\}$ for the Illness dataset and $T \in \{96, 192, 336, 720\}$ for all other datasets. Performance metrics for baseline models are obtained from Goswami et al. (2024), while our model's results are newly computed. A look-back window of $L = 512$ is used for all models, except DLinear, which employs an optimized input length of $L = 96$ to enhance performance.

### 4.2 MAIN RESULTS

Comprehensive forecasting results are listed in Table 1, with the best performance highlighted in **red** and the second best underlined. All model results are from (Goswami et al., 2024), except S-mamba and iTransformer, which we trained due to unavailable results for window size 512. A lower Mean Squared Error (MSE) and Mean Absolute Error (MAE) indicate more accurate predictions. Our proposed model, ParallelTime, demonstrates exceptional performance across a diverse set of datasets and prediction horizons, consistently outperforming a range of state-of-the-art models, ParallelTime achieves the best forecasting accuracy in a significant number of scenarios, particularly excelling in datasets such as Weather, ETTh1, ETTh2, ETTm2, Electricity, Traffic, and Illness.

Our model, ParallelTime, surpasses SOTA models, including PatchTST (Nie et al., 2023) and Moment (Goswami et al., 2024), in long-term time series forecasting. Although PatchTST remains a strong contender, ranking as the second-best performer, and Moment excels on the ETTm2 dataset (likely due to its training data), ParallelTime achieves superior performance with significantly fewer parameters and lower computational complexity, compared to PatchTST (see Table 2). Specifically, ParallelTime reduces MSE by an average of 4.25% and MAE by 4.31% relative to PatchTST. This combination of high accuracy, computational efficiency, and reduced resource requirements highlights the versatility and effectiveness

| Methods | ParallelTime | | S-Mamba | | iTransformer | | PatchTST | | DLinear | | TimesNet | | FEDFormer | | MOMENT | | GPT4TS | |
|---|---|---|---|---|---|---|---|---|---|---|---|---|---|---|---|---|---|---|
| Metric | MSE | MAE | MSE | MAE | MSE | MAE | MSE | MAE | MSE | MAE | MSE | MAE | MSE | MAE | MSE | MAE | MSE | MAE |
| **Weather** 96 | 0.145 | 0.189 | 0.158 | 0.210 | 0.168 | 0.219 | 0.149 | 0.198 | 0.176 | 0.237 | 0.172 | 0.220 | 0.217 | 0.296 | 0.154 | 0.209 | 0.162 | 0.212 |
| 192 | 0.189 | 0.232 | 0.203 | 0.252 | 0.211 | 0.255 | 0.194 | 0.241 | 0.220 | 0.282 | 0.219 | 0.261 | 0.276 | 0.336 | 0.197 | 0.248 | 0.204 | 0.248 |
| 336 | 0.242 | 0.273 | 0.258 | 0.292 | 0.260 | 0.292 | 0.245 | 0.282 | 0.265 | 0.319 | 0.280 | 0.306 | 0.339 | 0.380 | 0.246 | 0.285 | 0.254 | 0.286 |
| 720 | 0.323 | 0.331 | 0.328 | 0.340 | 0.332 | 0.341 | 0.314 | 0.334 | 0.333 | 0.362 | 0.365 | 0.359 | 0.403 | 0.428 | 0.315 | 0.336 | 0.326 | 0.337 |
| **ETTh1** 96 | 0.365 | 0.398 | 0.395 | 0.422 | 0.407 | 0.428 | 0.370 | 0.399 | 0.375 | 0.399 | 0.384 | 0.402 | 0.376 | 0.419 | 0.387 | 0.410 | 0.376 | 0.397 |
| 192 | 0.399 | 0.415 | 0.427 | 0.443 | 0.427 | 0.443 | 0.413 | 0.421 | 0.405 | 0.416 | 0.436 | 0.429 | 0.420 | 0.448 | 0.410 | 0.426 | 0.416 | 0.418 |
| 336 | 0.385 | 0.414 | 0.462 | 0.469 | 0.456 | 0.463 | 0.422 | 0.436 | 0.439 | 0.443 | 0.491 | 0.469 | 0.459 | 0.465 | 0.422 | 0.437 | 0.442 | 0.433 |
| 720 | 0.420 | 0.443 | 0.522 | 0.518 | 0.468 | 0.472 | 0.447 | 0.466 | 0.472 | 0.490 | 0.521 | 0.500 | 0.506 | 0.507 | 0.454 | 0.472 | 0.477 | 0.456 |
| **ETTh2** 96 | 0.262 | 0.328 | 0.298 | 0.356 | 0.298 | 0.357 | 0.274 | 0.336 | 0.289 | 0.353 | 0.340 | 0.374 | 0.358 | 0.397 | 0.288 | 0.345 | 0.285 | 0.342 |
| 192 | 0.322 | 0.368 | 0.372 | 0.399 | 0.377 | 0.406 | 0.339 | 0.379 | 0.383 | 0.418 | 0.402 | 0.414 | 0.429 | 0.439 | 0.349 | 0.386 | 0.354 | 0.389 |
| 336 | 0.312 | 0.370 | 0.402 | 0.432 | 0.424 | 0.440 | 0.329 | 0.380 | 0.448 | 0.465 | 0.452 | 0.452 | 0.496 | 0.487 | 0.369 | 0.408 | 0.373 | 0.407 |
| 720 | 0.399 | 0.434 | 0.419 | 0.449 | 0.438 | 0.462 | 0.379 | 0.422 | 0.605 | 0.551 | 0.462 | 0.468 | 0.463 | 0.474 | 0.403 | 0.439 | 0.406 | 0.441 |
| **ETTm1** 96 | 0.284 | 0.337 | 0.309 | 0.361 | 0.313 | 0.367 | 0.290 | 0.342 | 0.299 | 0.343 | 0.338 | 0.375 | 0.379 | 0.419 | 0.293 | 0.349 | 0.292 | 0.346 |
| 192 | 0.329 | 0.366 | 0.345 | 0.384 | 0.348 | 0.385 | 0.332 | 0.369 | 0.335 | 0.365 | 0.374 | 0.387 | 0.426 | 0.441 | 0.326 | 0.368 | 0.332 | 0.372 |
| 336 | 0.365 | 0.391 | 0.375 | 0.403 | 0.377 | 0.403 | 0.366 | 0.392 | 0.369 | 0.386 | 0.410 | 0.411 | 0.445 | 0.459 | 0.352 | 0.384 | 0.366 | 0.394 |
| 720 | 0.424 | 0.430 | 0.435 | 0.440 | 0.438 | 0.438 | 0.416 | 0.420 | 0.425 | 0.421 | 0.478 | 0.450 | 0.543 | 0.490 | 0.405 | 0.416 | 0.417 | 0.421 |
| **ETTm2** 96 | 0.162 | 0.252 | 0.177 | 0.270 | 0.179 | 0.273 | 0.165 | 0.255 | 0.167 | 0.269 | 0.187 | 0.267 | 0.203 | 0.287 | 0.170 | 0.260 | 0.173 | 0.262 |
| 192 | 0.218 | 0.291 | 0.229 | 0.305 | 0.242 | 0.315 | 0.220 | 0.292 | 0.224 | 0.303 | 0.249 | 0.309 | 0.269 | 0.328 | 0.227 | 0.297 | 0.229 | 0.301 |
| 336 | 0.276 | 0.327 | 0.281 | 0.338 | 0.291 | 0.345 | 0.274 | 0.329 | 0.281 | 0.342 | 0.321 | 0.351 | 0.325 | 0.366 | 0.275 | 0.328 | 0.286 | 0.341 |
| 720 | 0.356 | 0.380 | 0.371 | 0.392 | 0.377 | 0.398 | 0.362 | 0.385 | 0.397 | 0.421 | 0.408 | 0.403 | 0.421 | 0.415 | 0.363 | 0.387 | 0.378 | 0.401 |
| **Illness** 24 | 1.166 | 0.657 | 1.918 | 0.847 | 1.960 | 0.952 | 1.319 | 0.754 | 2.215 | 1.081 | 2.317 | 0.934 | 3.228 | 1.260 | 2.728 | 1.114 | 2.063 | 0.881 |
| 36 | 1.293 | 0.727 | 2.006 | 0.944 | 2.264 | 0.978 | 1.430 | 0.834 | 1.963 | 0.963 | 1.972 | 0.920 | 2.679 | 1.080 | 2.669 | 1.092 | 1.868 | 0.892 |
| 48 | 1.399 | 0.772 | 2.080 | 0.898 | 2.266 | 1.042 | 1.553 | 0.815 | 2.130 | 1.024 | 2.238 | 0.940 | 2.622 | 1.078 | 2.728 | 1.098 | 1.790 | 0.884 |
| 60 | 1.615 | 0.844 | 2.414 | 1.094 | 2.541 | 1.108 | 1.470 | 0.788 | 2.368 | 1.096 | 2.027 | 0.928 | 2.857 | 1.157 | 2.883 | 1.126 | 1.979 | 0.957 |
| **Electricity** 96 | 0.128 | 0.222 | 0.133 | 0.230 | 0.131 | 0.227 | 0.129 | 0.222 | 0.140 | 0.237 | 0.168 | 0.272 | 0.193 | 0.308 | 0.136 | 0.233 | 0.139 | 0.238 |
| 192 | 0.148 | 0.240 | 0.155 | 0.250 | 0.153 | 0.249 | 0.157 | 0.240 | 0.153 | 0.249 | 0.184 | 0.289 | 0.201 | 0.315 | 0.152 | 0.247 | 0.153 | 0.251 |
| 336 | 0.163 | 0.258 | 0.169 | 0.268 | 0.168 | 0.264 | 0.163 | 0.259 | 0.169 | 0.267 | 0.198 | 0.300 | 0.214 | 0.329 | 0.167 | 0.264 | 0.169 | 0.266 |
| 720 | 0.197 | 0.288 | 0.197 | 0.293 | 0.198 | 0.291 | 0.197 | 0.290 | 0.203 | 0.301 | 0.220 | 0.320 | 0.246 | 0.355 | 0.205 | 0.295 | 0.206 | 0.297 |
| **Traffic** 96 | 0.349 | 0.231 | 0.354 | 0.252 | 0.350 | 0.257 | 0.360 | 0.249 | 0.410 | 0.282 | 0.593 | 0.321 | 0.587 | 0.366 | 0.391 | 0.282 | 0.388 | 0.282 |
| 192 | 0.371 | 0.240 | 0.373 | 0.260 | 0.387 | 0.276 | 0.379 | 0.256 | 0.423 | 0.287 | 0.617 | 0.336 | 0.604 | 0.373 | 0.404 | 0.287 | 0.407 | 0.290 |
| 336 | 0.388 | 0.250 | 0.390 | 0.265 | 0.407 | 0.289 | 0.392 | 0.264 | 0.436 | 0.296 | 0.629 | 0.336 | 0.621 | 0.383 | 0.414 | 0.292 | 0.412 | 0.294 |
| 720 | 0.429 | 0.274 | 0.430 | 0.288 | 0.433 | 0.297 | 0.432 | 0.286 | 0.466 | 0.315 | 0.640 | 0.350 | 0.626 | 0.382 | 0.450 | 0.310 | 0.450 | 0.312 |

Table 1: The complete results of in-domain forecasting experiments. A lower MSE or MAE indicates a better prediction. Red: the best, Underline: the 2nd best.

Table 2: Comparison of ParallelTime and PatchTST on the Traffic dataset. The table reports MSE, MAE, forward and backward (Fwd+Bwd) FLOPs (i.e., training FLOPs), and the number of parameters (#Params). Bold values indicate superior performance. ↓ indicates that lower values are better. The improvement percentages for ParallelTime over PatchTST are shown in parentheses.

| | MSE | | MAE | | Fwd+Bwd FLOPs | | #Params | |
|---|---|---|---|---|---|---|---|---|
| Pred Len | **ParallelTime** | PatchTST | **ParallelTime** | PatchTST | **ParallelTime** | PatchTST | **ParallelTime** | PatchTST |
| 96 | **0.349 (↓3.1%)** | 0.360 | **0.231 (↓7.2%)** | 0.249 | **25.2G (↓36%)** | 39.5G | **614k (↓48%)** | 1194k |
| 192 | **0.371 (↓2.1%)** | 0.379 | **0.240 (↓6.3%)** | 0.256 | **25.2G (↓37%)** | 40.5G | **651k (↓67%)** | 1980k |
| 336 | **0.388 (↓1.0%)** | 0.392 | **0.250 (↓5.3%)** | 0.264 | **25.3G (↓39%)** | 42.1G | **707k (↓77%)** | 3160k |
| 720 | **0.429 (↓0.7%)** | 0.432 | **0.274 (↓4.2%)** | 0.286 | **25.5G (↓44%)** | 46.3G | **855k (↓86%)** | 6306k |

of ParallelTime, positioning it as a leading solution for real-world time series forecasting challenges.

# 5 MODEL ANALYSIS

## 5.1 PATCH-LEVEL WEIGHT ANALYSIS

To illustrate how our model allocates short-term and long-term dependencies for each token (patch), we analyze a sample from the Traffic dataset at prediction lengths of 96 and 192. We extract the weights assigned by our ParallelTime Weighter and present them in Figure 4. Looking at the input and the first block at each prediction length, when the previous patch (from left to right) exhibits a high value, our model assigns greater weight to Mamba, prioritizing long-term dependencies to reduce overfitting to potential noise. Similarly, in the

second block for each prediction length, when consecutive patches are similar, the model leverages Mamba to emphasize long-term dependencies, capturing a broader range of historical behaviors rather than focusing solely on recent patterns, facilitating precise predictions at the mutation boundary. Conversely, when preceding patches differ significantly, the model assigns more weight to the attention mechanism to prioritize short-term dependencies. Notably, for the second blocks, longer prediction lengths exhibit a stronger emphasis on long-term dependencies. For an additional result, see Appendix 9.3.

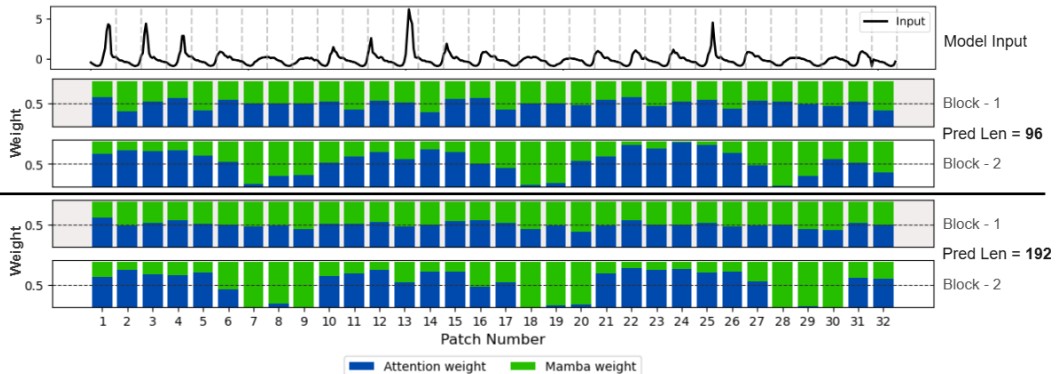

Figure 4: Visualization of input series and the weight distribution for prediction length 96, 192 per patch in sample from Traffic dataset, for each of the first and second ParallelTime blocks.

## 5.2 Dynamic Weighting Analysis

To evaluate the performance of our dynamic weighting mechanism across various datasets, we computed the mean weight of all tokens (patches) for each layer in our ParallelTime, as shown in Figure 5. The analysis includes the Weather, Electricity, ETTh1, and Traffic datasets.

The results demonstrate that, in the setting where the Attention-Mamba weights of each patch are averaged across all patches, each dataset emphasizes a different balance between short-term and long-term dependencies. Notably, across all datasets, the second layer consistently assigns more weight to the window attention mechanism compared to the first layer. For example, in the Weather dataset, when the prediction lengths are 192 and 336, the model relies more heavily on long-term dependencies, which are captured by the Mamba mechanism in the first layer. Conversely, for prediction lengths of 96 and 720, short-term dependencies are prioritized via the attention mechanism. In the second layer, attention receives a larger share of the weights regardless of the prediction length.

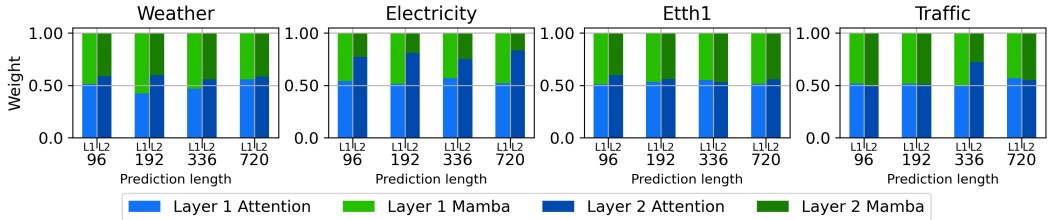

Figure 5: Mean weight of tokens (patches) per layer in the ParallelTime model, highlighting varying requirements for short-term and long-term dependencies across different datasets and prediction horizons.

# 6 ABLATION STUDY

## 6.1 WEIGHTING STRATEGY FOR ATTENTION AND MAMBA

We assess the impact of our proposed ParallelTime weighting methodology. We compare multiple strategies, including mean weighting, as in (Dong et al., 2024), and sum weighting. To ensure compatibility, Attention and Mamba outputs are normalized prior to weighting to address their differing scales. Our results, as shown in Figure 6, confirm the effectiveness of this approach across all datasets. Additional results for other datasets are provided in Appendix 9.1.

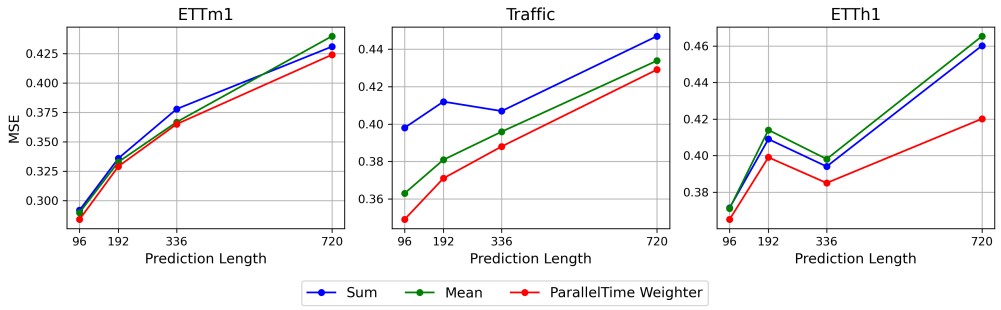

Figure 6: Ablation study of various weighting strategies - Mean, Sum and our ParallelTime Weighter for combining Attention and Mamba outputs.

## 6.2 MODEL EFFICIENCY ANALYSIS

Table 2 presents a comparison of MSE and MAE, Floating-Point Operations (FLOPs), and number of parameters, of our model against PatchTST across various prediction lengths using the Traffic dataset. The results show that our model requires significantly fewer FLOPs for both training and inference, achieves higher accuracy, and scales better with larger prediction lengths. This efficiency makes our model particularly well-suited for real-time long-term forecasting applications, where computational resources and speed are critical. For results on additional datasets, refer to Appendix 8.

# 7 CONCLUSION AND FUTURE WORK

In this work, we present ParallelTime, a novel decoder-only architecture that integrates local window attention and Mamba in parallel to effectively capture short-term and long-term dependencies, respectively. The outputs of these components are processed by our noval ParallelTime Weighter, which adaptively assigns weights to each component for accurate predictions. Our approach achieves state-of-the-art performance across multiple real-world benchmarks while requiring fewer parameters and lower computational costs. This work establishes a foundation for future advancements in parallel Attention-Mamba architectures, poised to enhance long-term time series forecasting.

Future research can explore the model's potential as a foundation for time series analysis with minimal adjustments. Specifically, efforts can focus on fine-tuning the model for diverse tasks, such as anomaly detection, classification, and multi-step forecasting, across various domains.

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

# 8 APPENDIX

## 8.1 DATASET

In this study, we assessed the efficacy of our approach by employing seven datasets widely recognized in the domain of long-term time series forecasting: Weather, Traffic, Electricity, Illness, and the ETT datasets (ETTh1, ETTh2, ETTm1, and ETTm2). These datasets encompass a diverse array of periodic patterns and real-world scenarios that present significant predictive challenges, rendering them particularly appropriate for applications such as long-term time series forecasting, data generation, and imputation tasks. The datasets are characterized by the following attributes: Dataset, Variants, Frequency, Timesteps, Information, Forecasting Horizon, and Term. Specifically, the Weather dataset comprises 21 meteorological variables recorded every 10 minutes at the Max Planck Biogeochemistry Institute's Weather Station in 2020. The Electricity dataset captures hourly electricity usage data from 321 customers. The Traffic dataset records hourly road occupancy rates from 862 sensors across San Francisco Bay Area freeways, spanning January 2015 to December 2016 (Zhou et al., 2021). The ETT datasets include 7 variables related to electricity transformers, collected from July 2016 to July 2018, consisting of four subsets: ETTh1 and ETTh2, recorded hourly, and ETTm1 and ETTm2, recorded every 15 minutes (Wu et al., 2021). The Illness dataset contains weekly data on patient numbers and influenza-like illness ratios (Nie et al., 2023). Detailed characteristics of these datasets are outlined in Table 3.

Table 3: Details of multivariate real-world datasets.

| Dataset | Variants | Timesteps | Information | Forecasting Horizon | Term |
|---|---|---|---|---|---|
| Weather | 21 | 52,696 | Weather | (96, 192, 336, 720) | 4 years |
| Electricity | 321 | 17,544 | Electricity | (96, 192, 336, 720) | 2 years |
| Traffic | 862 | 26,304 | Road occupancy | (96, 192, 336, 720) | - |
| Illness | 7 | 967 | health outcomes | (24, 36, 48, 60) | - |
| ETTh1 | 7 | 17,420 | electricity transformers | (96, 192, 336, 720) | 2 years |
| ETTh2 | 7 | 17,420 | electricity transformers | (96, 192, 336, 720) | 2 years |
| ETTm1 | 7 | 69,680 | electricity transformers | (96, 192, 336, 720) | 2 years |
| ETTm2 | 7 | 69,680 | electricity transformers | (96, 192, 336, 720) | 2 years |

## 8.2 TRAINING DETAILS AND HYPERPARAMETER SETTINGS

### 8.2.1 TRAINING

All our training was conducted on a single Nvidia RTX 4090. For optimization, we used the Adam optimizer (Kingma and Ba, 2015), which provides efficient adaptive learning rate adjustments. For the loss function to train the model, we used the classical Huber loss

function, chosen for its enhanced robustness to outliers and contribution to improved training stability.

**Efficient Training Strategy.** Given the extensive variety in datasets such as Electricity and Traffic, our model encounters memory constraints, even with small batch sizes, on the experimental hardware. Training on high-dimensional multivariate time series, common in real-world applications, is resource-intensive. To mitigate this, we adopt an efficient training strategy inspired by (Liu et al., 2024). Specifically, we randomly select a subset of variates for each batch, training the model exclusively on these variates to improve efficiency. For the Electricity and Traffic datasets, we use 30 randomly selected variates for the training set and 40 for the validation set, while the test set is used in its entirety.

### 8.2.2 Hyperparameter settings

We detail the hyperparameters employed in our ParallelTime model for long-term time series forecasting. These include common hyperparameters, applied uniformly across all datasets, and dataset-specific hyperparameters. Common settings include a random seed of 2023 for reproducibility, an input sequence length of 512, Huber loss with a delta of 1.0, attention dropout of 0.1, projection dropout of 0.05, 2 block layers with an attention head size of 4, a patch length of 16, a window attention length of 4, 32 register tokens, and Mamba settings with a state dimension of 16 and convolution dimension of 2. Dataset-specific settings in the table 4.

**More Details**: We have not explored optimizers beyond Adam. The attention mechanism utilized Flash Attention. We tested Absolute Positional Embedding, Rotary Positional Embedding, and Relative Positional Embedding, with Absolute Positional Embedding performing best.

Table 4: Hyperparameters for the ParallelTime model

| Parameter | Electricity | ETTh1 | ETTh2 | ETTm1 | ETTm2 | Illness | Traffic | Weather |
|---|---|---|---|---|---|---|---|---|
| epochs | 20 | 20 | 15 | 30 | 25 | 10 | 25 | 25 |
| lr | 0.005 | 0.0008 | 0.0006 | 0.0001 | 0.0001 | 0.012 | 0.005 | 0.0004 |
| batch | 64 | 256 | 512 | 64 | 512 | 64 | 64 | 64 |
| dim | 128 | 16 | 16 | 32 | 32 | 32 | 128 | 16 |

### 8.3 Component Selection

**Linear-Conv1D Embedding.** The proposed embedding method, designed to capture both global and local features, demonstrates modest performance improvements across most data sets. More research is required to fully understand the potential of this component and optimize its effectiveness.

**Global Registers.** The integration of global registers yields slight performance enhancements. We keep them because we believe that when scaling the model to a larger number of parameters, the model's performance can benefit.

**S4 vs. Mamba.** In our very original and clear paper, we did not choose to use S4 (Gu et al., 2022) instead of Mamba due to the limitations of S4, which exhibits deficiencies in the selective copying task and the induction heads task (Gu and Dao, 2023).

## 9 Additional results

### 9.1 Weighting Strategy

This section presents additional results for the weighting strategies applied to Attention and Mamba models, as discussed in Subsection 6.1. The findings demonstrate that, across all

prediction lengths and datasets, our ParallelTime Weighter consistently outperforms other weighting strategies, achieving the best results on every dataset.

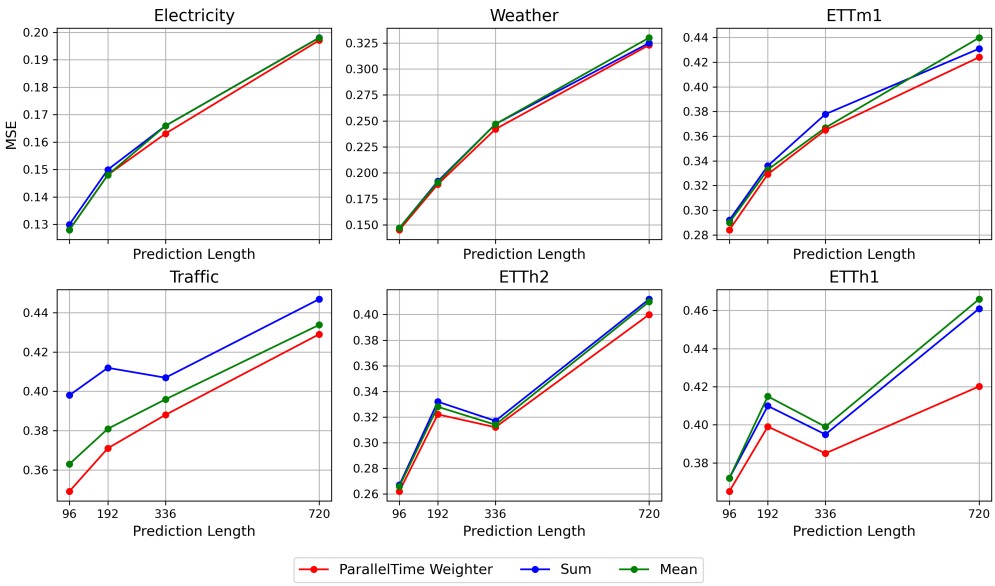

Figure 7: Performance comparison of weighting strategies for Attention and Mamba models across various prediction lengths and datasets, highlighting the superior results of our ParallelTime Weighter.

## 9.2 STUDY OF EXPAND-COMPRESS-PROJECT

In this subsection, we present a comparative analysis of our proposed Expand-Compress-Project method against the standard projection method in time series forecasting. Table 5 provides a detailed comparison across various datasets and prediction lengths. It is evident from the table that the our Expand-Compress-Project method consistently achieves similar and sometimes better MSE values to the standard projection method while significantly reduces the number of parameters required. In addition we can see that our model scales better on larger sequence length.

Table 5: Comparison of Expand-Compress-Project and Standard Projection Methods for different prediction lengths, where our method utilizes significantly fewer parameters while maintaining similar accuracy

| Dataset | | MSE | | #params | |
|---|---|---|---|---|---|
| | | Expand-Compress | Standard Projection | Standard Projection | Expand-Compress |
| Electricity | 96 | 0.128 | 0.127 | 854.432 K | **516 K** |
| | 192 | 0.148 | 0.146 | 1.2477 M | **552.96 K** |
| | 336 | 0.163 | 0.162 | 1.8377 M | **608.4 K** |
| | 720 | 0.197 | 0.196 | 3.411 M | **756.24 K** |
| Traffic | 96 | 0.349 | 0.353 | 953.248 K | **614.816 K** |
| | 192 | 0.371 | 0.372 | 1.3466 M | **651.776 K** |
| | 336 | 0.389 | 0.389 | 1.9365 M | **707.216 K** |
| | 720 | 0.430 | 0.432 | 3.5098 M | **855.056 K** |

## 9.3 Patch-Level Weight Additional Analysis

We visualize a sample from the ETTM1 dataset to illustrate the distribution of short-term and long-term dependencies utilized by our model for each token (patch). We extract the weights assigned by our ParallelTime Weighter and present them in Figure 8. The visualization reveals that patches significantly different from preceding patches (from left to right) rely more heavily on the Mamba weights, which emphasize long-term dependencies. Conversely, when the data exhibits minimal variation, greater weight is assigned to window attention, which prioritizes short-term dependencies. Additionally, we observe distinct behaviors across different layers.

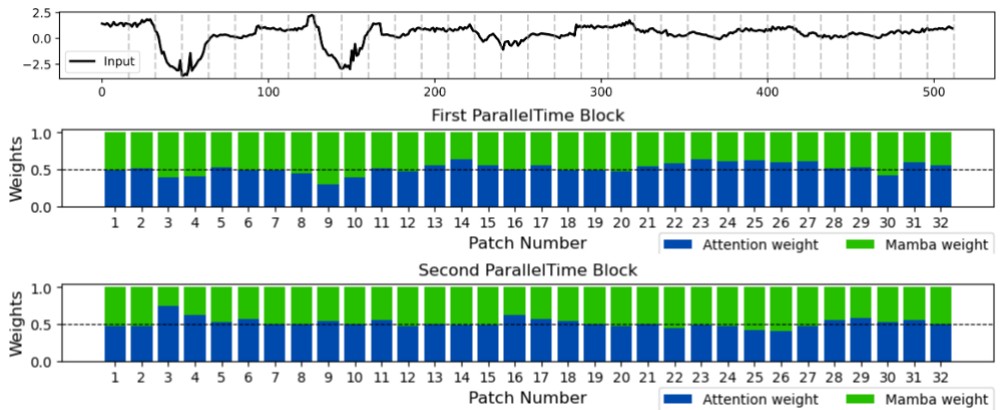

Figure 8: Visualization of input series and the weight distribution per patch in sample from ETTM1 dataset, for the first and second ParallelTime blocks.

## 9.4 Comparison of ParallelTime, Pure Attention, and Pure Mamba Blocks

To further assess the contribution of the ParallelTime block, we conduct an ablation study in which we retrain the model after replacing this block with either an attention-only or a Mamba-only variant. The results are summarized in Table 6. Our model consistently achieves superior performance on datasets with large training horizons—namely Electricity, Traffic, and Weather—across nearly all prediction lengths. For the ETT datasets, which contain relatively short and less dense sequences, the differences between methods are smaller and the outcomes more mixed. This strengthens our suggestion that the ParallelTime Weighter is better suited for datasets with more data.

## 9.5 Robustness

**Effects of Different Parameter Adjustments.** To evaluate the impact of hyperparameter choices on ParallelTime, we conducted additional experiments by adjusting key model parameters. We tested different configurations by varying the number of ParallelTime layers, $L = 1, 2, 3$, and the patch size, $P = 8, 16$, resulting in a total of six unique hyperparameter combinations. The MSE scores for these configurations across various datasets are presented in Figure 9. Most datasets show consistent performance across hyperparameter settings, except for the ILI dataset, which exhibits slightly variable results.

**Impact of Various Random Seeds.** The findings presented in the main text and appendix were obtained using a consistent random seed of 2023. To assess the stability of these outcomes, we trained the supervised ParallelTime model using five random seeds: 2022, 2023, 2024, 2025, and 2026, computing the MSE and MAE scores for each seed. The average and standard deviation of these results are shown in Table 9.5. The notably low standard deviations demonstrate that our model's performance remains stable across different random seed selections.

Table 6: Comparison of ParallelTime, Attention, and Mamba, **bold** is best value.

| Dataset | Length | ParallelTime | | Attention | | Mamba | |
| --- | --- | --- | --- | --- | --- | --- | --- |
| | | MSE | MAE | MSE | MAE | MSE | MAE |
| Electricity | 96 | **0.127** | **0.222** | 0.128 | 0.222 | 0.129 | 0.224 |
| | 192 | 0.147 | **0.241** | 0.148 | 0.241 | **0.146** | 0.242 |
| | 336 | **0.163** | **0.258** | 0.166 | 0.259 | 0.163 | 0.259 |
| | 720 | **0.197** | **0.288** | 0.199 | 0.290 | 0.199 | 0.291 |
| Traffic | 96 | **0.349** | **0.231** | 0.353 | 0.233 | 0.364 | 0.244 |
| | 192 | **0.371** | **0.240** | 0.373 | 0.243 | 0.383 | 0.251 |
| | 336 | **0.388** | **0.250** | 0.388 | 0.253 | 0.391 | 0.256 |
| | 720 | **0.429** | **0.274** | 0.430 | 0.276 | 0.436 | 0.279 |
| Weather | 96 | **0.145** | **0.189** | 0.146 | 0.191 | 0.149 | 0.194 |
| | 192 | **0.189** | **0.232** | 0.191 | 0.234 | 0.193 | 0.235 |
| | 336 | **0.242** | **0.273** | 0.244 | 0.275 | 0.243 | 0.274 |
| | 720 | 0.323 | 0.331 | **0.317** | **0.329** | 0.320 | 0.330 |
| ETTh1 | 96 | 0.365 | 0.398 | 0.368 | 0.398 | **0.363** | **0.395** |
| | 192 | **0.399** | 0.415 | 0.413 | 0.427 | 0.399 | **0.413** |
| | 336 | **0.385** | **0.414** | 0.407 | 0.432 | 0.388 | 0.418 |
| | 720 | **0.420** | **0.443** | 0.454 | 0.469 | 0.451 | 0.464 |
| ETTh2 | 96 | **0.262** | **0.328** | 0.263 | 0.328 | 0.264 | 0.329 |
| | 192 | **0.322** | 0.368 | 0.324 | 0.368 | 0.322 | **0.367** |
| | 336 | **0.312** | **0.370** | 0.314 | 0.371 | 0.318 | 0.373 |
| | 720 | 0.399 | 0.434 | **0.397** | **0.433** | 0.405 | 0.438 |
| ETTm1 | 96 | **0.284** | **0.337** | 0.286 | 0.341 | 0.291 | 0.339 |
| | 192 | 0.329 | 0.366 | **0.327** | 0.366 | 0.329 | **0.363** |
| | 336 | 0.365 | 0.391 | **0.354** | 0.384 | 0.359 | **0.383** |
| | 720 | **0.424** | 0.430 | 0.425 | 0.427 | 0.430 | **0.423** |
| ETTm2 | 96 | **0.162** | **0.252** | 0.163 | 0.252 | 0.163 | 0.253 |
| | 192 | 0.218 | 0.291 | 0.219 | **0.290** | **0.216** | 0.290 |
| | 336 | 0.276 | 0.327 | 0.273 | 0.324 | **0.270** | **0.323** |
| | 720 | 0.356 | 0.380 | 0.355 | 0.379 | **0.350** | **0.374** |

**Robustness to Window Size.** We demonstrate that our model is robust to different window sizes. We evaluated window sizes of 32, 64, and 128, and report the corresponding mean and standard deviation for each configuration in Table 9.5.

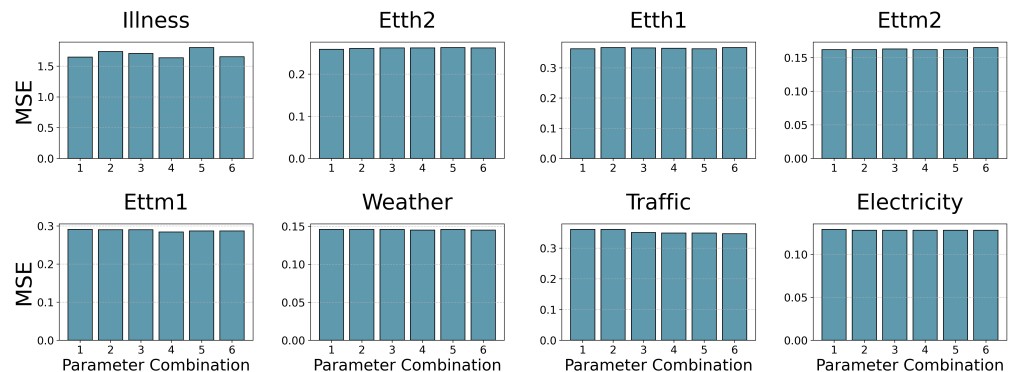

Figure 9: MSE scores for ParallelTime across six hyperparameter configurations (number of layers $L = 1, 2, 3$, patch size $P = 8, 16$).

| Dataset | ETTh2 | | Traffic | | Weather | |
|---------|-------|-------|---------|-------|---------|-------|
| Horizon | MSE | MAE | MSE | MAE | MSE | MAE |
| 96 | 0.263±0.0011 | 0.328±0.0007 | 0.350±0.0009 | 0.231±0.0000 | 0.146±0.0007 | 0.190±0.0011 |
| 192 | 0.323±0.0011 | 0.368±0.0013 | 0.371±0.0000 | 0.241±0.0005 | 0.191±0.0011 | 0.234±0.0011 |
| 336 | 0.313±0.0008 | 0.371±0.0013 | 0.390±0.0012 | 0.252±0.0011 | 0.244±0.0015 | 0.276±0.0015 |
| 720 | 0.404±0.0036 | 0.437±0.0027 | 0.429±0.0009 | 0.274±0.0004 | 0.324±0.0026 | 0.331±0.0015 |

| Dataset | ETTm1 | | ETTm2 | | Electricity | |
|---------|-------|-------|-------|-------|-------------|-------|
| Horizon | MSE | MAE | MSE | MAE | MSE | MAE |
| 96 | 0.289±0.0036 | 0.341±0.0035 | 0.162±0.0004 | 0.252±0.0004 | 0.128±0.0004 | 0.222±0.0004 |
| 192 | 0.330±0.0019 | 0.368±0.0021 | 0.221±0.0029 | 0.292±0.0016 | 0.147±0.0005 | 0.240±0.0015 |
| 336 | 0.361±0.0025 | 0.389±0.0012 | 0.276±0.0023 | 0.327±0.0011 | 0.164±0.0008 | 0.258±0.0004 |
| 720 | 0.436±0.0085 | 0.434±0.0034 | 0.356±0.0046 | 0.380±0.0022 | 0.197±0.0008 | 0.288±0.0010 |

Table 7: Robustness from five different random seeds.

| Dataset | Pred Len | MSE | | MAE | | Fwd FLOPs | | Fwd+Bwd FLOPs | | #Params | |
|---------|----------|-----|---|-----|---|-----------|---|---------------|---|---------|---|
| | | ParallelTime | PatchTST | ParallelTime | PatchTST | ParallelTime | PatchTST | ParallelTime | PatchTST | ParallelTime | PatchTST |
| ETTh1 | 96 | **0.365** (↓1.4%) | 0.370 | **0.398** (↓0.3%) | 0.399 | **0.325G** (↓52%) | 0.687G | **0.976G** (↓52%) | 2.062G | **69k** (↓40%) | 116k |
| | 192 | **0.399** (↓3.4%) | 0.413 | **0.415** (↓1.4%) | 0.421 | **0.347G** (↓52%) | 0.731G | **1.042G** (↓52%) | 2.194G | **119k** (↓44%) | 214k |
| | 336 | **0.385** (↓8.8%) | 0.422 | **0.414** (↓5.0%) | 0.436 | **0.380G** (↓52%) | 0.797G | **1.141G** (↓52%) | 2.392G | **192k** (↓46%) | 362k |
| | 720 | **0.420** (↓6.0%) | 0.447 | **0.443** (↓4.9%) | 0.466 | **0.468G** (↓51%) | 0.973G | **1.405G** (↓51%) | 2.920G | **389k** (↓48%) | 755k |
| Electricity | 96 | **0.128** (↓0.8%) | 0.129 | **0.222** | 0.222 | **7.00G** (↓47%) | 13.2G | **21.0G** (↓47%) | 39.5G | **516k** (↓57%) | 1194k |
| | 192 | **0.148** (↓5.7%) | 0.157 | 0.241 | **0.240** | **7.02G** (↓48%) | 13.5G | **21.1G** (↓48%) | 40.6G | **553k** (↓72%) | 1981k |
| | 336 | **0.163** | 0.163 | **0.258** (↓0.4%) | 0.259 | **7.04G** (↓50%) | 14.1G | **21.1G** (↓50%) | 42.2G | **608k** (↓81%) | 3161k |
| | 720 | **0.196** (↓0.5%) | 0.197 | **0.288** (↓0.7%) | 0.290 | **7.11G** (↓54%) | 15.5G | **21.3G** (↓54%) | 46.4G | **756k** (↓88%) | 6307k |

Table 8: Comparison of ParallelTime and PatchTST on ETTh1 and Electricity datasets. The table reports MSE, MAE, forward (Fwd) FLOPs (i.e., inference FLOPs), forward and backward (Fwd+Bwd) FLOPs (i.e., training FLOPs), and the number of parameters (#Params) for different prediction lengths (Pred Len).

| Dataset | ETTh1 | | ETTh2 | | ETTm1 | |
|---------|-------|-----|-------|-----|-------|-----|
| Horizon | MSE | MAE | MSE | MAE | MSE | MAE |
| 96 | 0.366±0.0002 | 0.398±0.0005 | 0.264±0.0012 | 0.329±0.0007 | 0.286±0.0011 | 0.339±0.0006 |
| 192 | 0.400±0.0002 | 0.416±0.0007 | 0.322±0.0009 | 0.368±0.0005 | 0.329±0.0011 | 0.367±0.0009 |
| 336 | 0.386±0.0018 | 0.416±0.0017 | 0.313±0.0006 | 0.371±0.0007 | 0.364±0.0011 | 0.391±0.0011 |
| 720 | 0.422±0.0019 | 0.445±0.0017 | 0.400±0.0006 | 0.435±0.0005 | 0.425±0.0033 | 0.430±0.0028 |

Table 9: Robustness to different window sizes (32, 64, 128).

| Dataset | ETTh1 | | ETTh2 | | ETTm1 | |
|---------|-------|-----|-------|-----|-------|-----|
| Horizon | MSE | MAE | MSE | MAE | MSE | MAE |
| 96 | 0.366±0.0002 | 0.398±0.0005 | 0.264±0.0012 | 0.329±0.0007 | 0.286±0.0011 | 0.339±0.0006 |
| 192 | 0.400±0.0002 | 0.416±0.0007 | 0.322±0.0009 | 0.368±0.0005 | 0.329±0.0011 | 0.367±0.0009 |
| 336 | 0.386±0.0018 | 0.416±0.0017 | 0.313±0.0006 | 0.371±0.0007 | 0.364±0.0011 | 0.391±0.0011 |
| 720 | 0.422±0.0019 | 0.445±0.0017 | 0.400±0.0006 | 0.435±0.0005 | 0.425±0.0033 | 0.430±0.0028 |
| Dataset | ETTm2 | | Electricity | | Traffic | |
| Horizon | MSE | MAE | MSE | MAE | MSE | MAE |
| 96 | 0.162±0.0002 | 0.253±0.0001 | 0.128±0.0002 | 0.222±0.0002 | 0.350±0.0005 | 0.231±0.0003 |
| 192 | 0.220±0.0011 | 0.291±0.0003 | 0.148±0.0003 | 0.241±0.0003 | 0.371±0.0004 | 0.240±0.0002 |
| 336 | 0.276±0.0001 | 0.328±0.0007 | 0.165±0.0008 | 0.259±0.0005 | 0.390±0.0011 | 0.251±0.0003 |
| 720 | 0.359±0.0021 | 0.381±0.0006 | 0.198±0.0006 | 0.289±0.0006 | 0.430±0.0002 | 0.274±0.0004 |
| Dataset | Weather | | | | | |
| Horizon | MSE | MAE | | | | |
| 96 | 0.145±0.0005 | 0.189±0.0004 | | | | |
| 192 | 0.190±0.0007 | 0.232±0.0006 | | | | |
| 336 | 0.244±0.0024 | 0.274±0.0018 | | | | |
| 720 | 0.325±0.0032 | 0.332±0.0011 | | | | |

Table 10: Robustness to different window sizes (32, 64, and 128) reported as mean and standard deviation.

