# OpenReview forum: "ParallelTime: Dynamically Weighting the Balance of Short- and Long-Term Temporal Dependencies"
_ICLR.cc/2026/Conference — Submitted to ICLR 2026_

### Official Review · Reviewer_oNcd · 2025-10-25

**Soundness:** 3
**Presentation:** 3
**Contribution:** 2
**Rating:** 4
**Confidence:** 5

**Summary:**

This paper introduces ParallelTime, a new architecture for time series forecasting. It addresses the suboptimal equal weighting of long- and short-term dependencies in existing hybrid models with a dynamic weighting mechanism (ParallelTime Weighter), achieving state-of-the-art performance with improved efficiency and robustness.

**Strengths:**

1. The proposed ParallelTime Weighter is interesting. It dynamically and interdependently assigns weights to short-term and long-term components for each token.

2. A significant advantage is the achievement of superior performance with notably fewer parameters and lower computational cost compared to strong baselines like PatchTST.

3. The model demonstrates SOTA or highly competitive results across eight real-world benchmarks, effectively validating the proposed architecture's effectiveness.

**Weaknesses:**

1. The analysis, particularly regarding the ParallelTime Weighter, remains somewhat superficial. It lacks a clear explanation of why and when the model chooses to emphasize short- vs. long-term dependencies based on specific time series patterns or characteristics.

2. While the integration is novel, several core components (patching, channel independence, window attention, Mamba, registers) are directly adopted from existing literature.

3. The ablation analysis is insufficient as it fails to deconstruct the model's core contributions. To strengthen the claims, performance should be compared against the following ablated versions: 1) Mamba-only, 2) Attention-only, 3) without Registers, etc.

**Questions:**

I noticed noticeable inconsistencies in the title and font formatting compared to the standard ICLR template. Could the authors confirm the technical compliance of their document with the submission guidelines?

---

> ### Author Response · Authors · 2025-11-21
> **point-by-point reply part 1**
>
> We extend our sincere thanks to reviewer oNcd for their insightful and constructive feedback, which has significantly strengthened the clarity and rigor of our manuscript.
>
> **Weakness**
>
> W1 (Why and when the Parallel Weighter chooses to give more weight to each component): In Figures 4 and 8, we observe that, across all datasets, the token preceding the mutation point receives a higher Mamba component (indicating a preference for longer dependencies). This shows how our model not only attends to the mutation itself but also to what happened in the past, which reduces the amount of attention at the mutation point. Additionally, Figure 5 illustrates that different datasets require varying amounts of short- and long-term dependencies.
>
> W2 (Integration is novel, but core components are adopted from existing literature): Our main component and contribution is the Parallel Weighter. Please see the ablation study below (W3)
>
>
> W3 (Add more ablations): Thank you for bringing this up. We have added your suggested ablation study to our paper, in which we retrained the model by replacing the ParallelTime block with no registers, attention-only, and Mamba-only variants. The results are shown below. Our model yields better results in all datasets where a large number of timestamps is used for training (electricity, traffic, weather). In the other dataset (ETT), where the data points are small, the results are mixed. In addition, we can see that the registers do not impact the final solution a lot but can benefit it sometimes (as we declared in the paper). As we state in our conclusion, we believe that the Parallel Weigher can be very successful as a foundation for time series models where the number of timestamps that have been fed to the model is large.
>
> **Table in the next comment below**

---

> > ### Author Response · Authors · 2025-11-21
> > **point-by-point reply part 2**
> >
> > W3 (ablation results):
> >
> > |                      |   ParallelTime |   ParallelTime |   ParallelTime_no_reg |   ParallelTime_no_reg |   attention |   attention | mamba |   mamba |
> > |:---------------------|--------------------------:|--------------------------:|---------------------------------:|---------------------------------:|-----------------------:|-----------------------:|-------------------:|-------------------:|
> > |                      |   **MSE** |   **MAE** |   **MSE** |   **MAE** |   **MSE** |   **MAE** | **MSE** |   **MAE** |
> > | ('electricity', 96)  | **0.127928** | 0.222099 | 0.128254 | **0.222022** | 0.128914 | 0.222445 | 0.129143 | 0.224978 |
> > | ('electricity', 192) | 0.147846 | 0.241042 | 0.148698 | **0.240909** | 0.148369 | 0.241108 | **0.146882** | 0.242145 |
> > | ('electricity', 336) | 0.163897 | **0.258374** | 0.164452 | 0.25853 | 0.166179 | 0.259776 | **0.163415** | 0.259596 |
> > | ('electricity', 720) | 0.197931 | 0.288972 | **0.1958** | **0.28777** | 0.19959 | 0.290493 | 0.199651 | 0.291929 |
> > | ('traffic', 96)      | 0.349841 | 0.231143 | **0.349444** | **0.230726** | 0.353563 | 0.233127 | 0.364035 | 0.244112 |
> > | ('traffic', 192)     | **0.371046** | **0.240327** | 0.371078 | 0.24116 | 0.373405 | 0.243209 | 0.383265 | 0.251615 |
> > | ('traffic', 336)     | 0.388667 | **0.250633** | 0.38973 | 0.252111 | **0.388387** | 0.253613 | 0.391563 | 0.256112 |
> > | ('traffic', 720)     | **0.429942** | **0.274369** | 0.430469 | 0.274832 | 0.43038 | 0.276113 | 0.436759 | 0.279574 |
> > | ('weather', 96)      | **0.145717** | **0.189662** | 0.145923 | 0.190524 | 0.146944 | 0.191586 | 0.149348 | 0.194281 |
> > | ('weather', 192)     | **0.189806** | **0.232382** | 0.191951 | 0.233995 | 0.191688 | 0.234239 | 0.193433 | 0.235974 |
> > | ('weather', 336)     | **0.242879** | **0.273035** | 0.244015 | 0.275675 | 0.244526 | 0.275252 | 0.243764 | 0.274535 |
> > | ('weather', 720)     | 0.323511 | 0.331001 | 0.322331 | 0.330474 | **0.317896** | **0.32952** | 0.320841 | 0.330499 |
> > | ('ETTh1', 96)        | 0.365645 | 0.398115 | 0.364264 | 0.397478 | 0.368454 | 0.398398 | **0.363999** | **0.395025** |
> > | ('ETTh1', 192)       | 0.39935 | 0.415497 | 0.40521 | 0.421061 | 0.413961 | 0.427931 | **0.399064** | **0.413634** |
> > | ('ETTh1', 336)       | **0.385301** | **0.414899** | 0.399703 | 0.426818 | 0.407235 | 0.43215 | 0.38875 | 0.418704 |
> > | ('ETTh1', 720)       | **0.420452** | **0.443927** | 0.437497 | 0.457225 | 0.454514 | 0.469012 | 0.451458 | 0.46429 |
> > | ('ETTh2', 96)        | 0.262745 | 0.328645 | **0.260408** | **0.325749** | 0.26351 | 0.328278 | 0.264874 | 0.329455 |
> > | ('ETTh2', 192)       | 0.322967 | 0.368358 | **0.318956** | **0.364202** | 0.324503 | 0.368115 | 0.322532 | 0.367442 |
> > | ('ETTh2', 336)       | **0.312345** | 0.370723 | 0.313402 | **0.370451** | 0.314258 | 0.371006 | 0.318166 | 0.373097 |
> > | ('ETTh2', 720)       | 0.399933 | 0.434072 | 0.409034 | 0.441729 | **0.397953** | **0.43328** | 0.405937 | 0.43807 |
> > | ('ETTm1', 96)        | **0.28487** | **0.337948** | 0.288122 | 0.341916 | 0.286231 | 0.341464 | 0.29106 | 0.33942 |
> > | ('ETTm1', 192)       | 0.329177 | 0.366065 | 0.330998 | 0.369284 | **0.327874** | 0.366504 | 0.329152 | **0.363234** |
> > | ('ETTm1', 336)       | 0.365161 | 0.391733 | 0.357275 | 0.38693 | **0.354601** | 0.384871 | 0.359449 | **0.383618** |
> > | ('ETTm1', 720)       | **0.424969** | 0.430906 | 0.436661 | 0.435038 | 0.425285 | 0.427482 | 0.430536 | **0.42393** |
> > | ('ETTm2', 96)        | **0.162267** | 0.252789 | 0.162634 | **0.252642** | 0.163287 | 0.25284 | 0.163105 | 0.253653 |
> > | ('ETTm2', 192)       | 0.21899 | 0.291039 | 0.219589 | 0.291408 | 0.219202 | **0.290193** | **0.216836** | 0.290547 |
> > | ('ETTm2', 336)       | 0.276228 | 0.327567 | 0.277604 | 0.328394 | 0.273177 | 0.324154 | **0.270519** | **0.323236** |
> > | ('ETTm2', 720)       | 0.35681 | 0.380052 | 0.356438 | 0.382286 | 0.355661 | 0.37955 | **0.350588** | **0.374567** |
> >
> >
> > Q1 (Is it the right format?): We use the original ICLR template for formatting our paper.
> >
> > We trust that these replies address any lingering questions and more effectively underscore the innovative aspects and robust empirical results of ParallelTime. With these additional insights, we respectfully ask that you revisit your assessment, confident that our contribution marks a significant step forward in scalable time series prediction.

---

> > > ### Comment · Reviewer_oNcd · 2025-11-22
> > > **Response to Authors**
> > >
> > > Thanks for the authors’ efforts in providing additional ablation studies. I decide to raise my score from 4 to 6.
> > >
> > > However, I still think the formatting style is not appropriate. You may consider comparing the style of your submission with that of other ICLR submissions. For example, there is usually a sentence like “Under review as a conference paper at ICLR 2026” on the top of paper, and the paper title is typically in bold font.

---

> > > > ### Author Response · Authors · 2025-12-01
> > > > **Thank you for the score upgrade**
> > > >
> > > > Thank you for your careful review and for raising your score after considering our additional experiments. We will ensured that our formatting fully complies with ICLR guidelines and appreciate your suggestion.

---

### Official Review · Reviewer_gvPa · 2025-10-29

**Soundness:** 2
**Presentation:** 2
**Contribution:** 2
**Rating:** 4
**Confidence:** 5

**Summary:**

This paper proposes a dynamic weighting mechanism to balance the time dependency between short and long cycles, thereby achieving better performance.

**Strengths:**

This paper uses local attention and Mamba to extract information at different time intervals and performs adaptive weighted ensemble.

**Weaknesses:**

1、The method is essentially a multi-scale framework that divides the time series into long-term and short-term paths for separate feature extraction, without introducing any new modeling mechanism.

2、The fusion of long- and short-term features relies on simple weighting or concatenation, lacking an adaptive interaction mechanism.

3、Moreover, it merely employs Mamba and Attention to extract their respective effective features, but contributes no architectural or algorithmic innovation.

4、There is no theoretical or empirical evidence showing that this parallel dual-branch design outperforms existing multi-scale or frequency-decomposition methods.

5、Overall, the improvement lies purely at the implementation level, without any new algorithmic principle or fundamental contribution.

**Questions:**

See Weaknesses.

---

> ### Author Response · Authors · 2025-11-21
>
> We deeply value Reviewer gvPa's incisive review
>
> W1, W3, and W5 (What is our main contribution?): Our Parallel Weigher, which dynamically selects the balance between short- and long-term dependencies, is novel in time series forecasting; we achieve SOTA accuracy while using a small number of parameters.
> W2 (Does Parallel Time just weight the long- and short-term dependencies?): Our weighting architecture does not just assign weights to each component; we calculate weights with respect to each dependency, similar to attention weighting but more akin to the Squeeze-and-Excitation approach. That is why our method is successful; for more details, please see Section 3.4.
> W4 (Is ParallelTime better empirically than other methods?): We show that our method outperforms all SOTA models, as presented in our paper, including attention models like PatchTST [1] and Mamba models like S-Mamba [2].
>
> We hope these responses alleviate any remaining concerns and better highlight the novelty and empirical strengths of ParallelTime. Given these clarifications, we kindly request that you reconsider your overall rating, as we believe our work represents a meaningful advancement in efficient time series forecasting.
>
> [1] A Time Series is Worth 64 Words: Long-term Forecasting with Transformers
> [2] Is Mamba Effective for Time Series Forecasting?

---

> ### Author Response · Authors · 2025-11-27
>
> Dear Reviewer gvPa,
>
> Thank you for your review.
> We show in Table 1 that our method substantially outperforms S-Mamba [1], one of the leading Mamba-based time series models, and PatchTST [2], one of the strongest attention-based models.
>
> In addition, we will include the following table, in which we retrain the model by replacing the ParallelTime block with attention-only and Mamba-only variants. The results are shown below. Our model yields better results in all datasets where a large number of timestamps is used for training (electricity, traffic, weather). In the other dataset (ETT), where the data points are small, the results are mixed. As we state in our conclusion, we believe that the Parallel Weigher can be very successful as a foundation for time series models. Thank you for bringing this up. We will include this table in our paper!
>
>
> |                      |   ParallelTime |   ParallelTime |   attention |   attention |   mamba |   mamba |
> |:---------------------|--------------------------:|--------------------------:|-----------------------:|-----------------------:|-------------------:|-------------------:|
> |                      |   MSE |   MAE |   MSE |   MAE |   MSE |   MAE |
> | ('electricity', 96)  | **0.127928** | **0.222099** | 0.128914 | 0.222445 | 0.129143 | 0.224978 |
> | ('electricity', 192) | 0.147846 | **0.241042** | 0.148369 | 0.241108 | **0.146882** | 0.242145 |
> | ('electricity', 336) | 0.163897 | **0.258374** | 0.166179 | 0.259776 | **0.163415** | 0.259596 |
> | ('electricity', 720) | **0.197931** | **0.288972** | 0.19959 | 0.290493 | 0.199651 | 0.291929 |
> | ('traffic', 96)      | **0.349841** | **0.231143** | 0.353563 | 0.233127 | 0.364035 | 0.244112 |
> | ('traffic', 192)     | **0.371046** | **0.240327** | 0.373405 | 0.243209 | 0.383265 | 0.251615 |
> | ('traffic', 336)     | 0.388667 | **0.250633** | **0.388387** | 0.253613 | 0.391563 | 0.256112 |
> | ('traffic', 720)     | **0.429942** | **0.274369** | 0.43038 | 0.276113 | 0.436759 | 0.279574 |
> | ('weather', 96)      | **0.145717** | **0.189662** | 0.146944 | 0.191586 | 0.149348 | 0.194281 |
> | ('weather', 192)     | **0.189806** | **0.232382** | 0.191688 | 0.234239 | 0.193433 | 0.235974 |
> | ('weather', 336)     | **0.242879** | **0.273035** | 0.244526 | 0.275252 | 0.243764 | 0.274535 |
> | ('weather', 720)     | 0.323511 | 0.331001 | **0.317896** | **0.32952** | 0.320841 | 0.330499 |
> | ('ETTh1', 96)        | 0.365645 | 0.398115 | 0.368454 | 0.398398 | **0.363999** | **0.395025** |
> | ('ETTh1', 192)       | 0.39935 | 0.415497 | 0.413961 | 0.427931 | **0.399064** | **0.413634** |
> | ('ETTh1', 336)       | **0.385301** | **0.414899** | 0.407235 | 0.43215 | 0.38875 | 0.418704 |
> | ('ETTh1', 720)       | **0.420452** | **0.443927** | 0.454514 | 0.469012 | 0.451458 | 0.46429 |
> | ('ETTh2', 96)        | **0.262745** | 0.328645 | 0.26351 | **0.328278** | 0.264874 | 0.329455 |
> | ('ETTh2', 192)       | 0.322967 | 0.368358 | 0.324503 | 0.368115 | **0.322532** | **0.367442** |
> | ('ETTh2', 336)       | **0.312345** | **0.370723** | 0.314258 | 0.371006 | 0.318166 | 0.373097 |
> | ('ETTh2', 720)       | 0.399933 | 0.434072 | **0.397953** | **0.43328** | 0.405937 | 0.43807 |
> | ('ETTm1', 96)        | **0.28487** | **0.337948** | 0.286231 | 0.341464 | 0.29106 | 0.33942 |
> | ('ETTm1', 192)       | 0.329177 | 0.366065 | **0.327874** | 0.366504 | 0.329152 | **0.363234** |
> | ('ETTm1', 336)       | 0.365161 | 0.391733 | **0.354601** | 0.384871 | 0.359449 | **0.383618** |
> | ('ETTm1', 720)       | **0.424969** | 0.430906 | 0.425285 | 0.427482 | 0.430536 | **0.42393** |
> | ('ETTm2', 96)        | **0.162267** | **0.252789** | 0.163287 | 0.25284 | 0.163105 | 0.253653 |
> | ('ETTm2', 192)       | 0.21899 | 0.291039 | 0.219202 | **0.290193** | **0.216836** | 0.290547 |
> | ('ETTm2', 336)       | 0.276228 | 0.327567 | 0.273177 | 0.324154 | **0.270519** | **0.323236** |
> | ('ETTm2', 720)       | 0.35681 | 0.380052 | 0.355661 | 0.37955 | **0.350588** | **0.374567** |
>
> Given this additional evidence, we would be very grateful if you would consider updating your score accordingly.
>
> Thanks, the authors.

---

### Official Review · Reviewer_vQqx · 2025-10-30

**Soundness:** 3
**Presentation:** 3
**Contribution:** 3
**Rating:** 4
**Confidence:** 3

**Summary:**

This paper proposed **ParallelTime** to combine two dominant architectures, Transformer and Mamba. While prior work in NLP has combined these models, it typically does so by simple averaging, which assigns equal weight to both short- and long-term features. The authors argue this is suboptimal for time series data.

The core contributions are:
1. **ParallelTime Architecture**: A decoder-only model that tokenizes the input multivariable time-series data as patchifed univariable time-series and processes them through a combination of Mamba blocks and window-attention blocks by **ParallelTime Weighter** instead of simply summing or averaging.
1. **ParallelTime Weighter**: A novel, dynamic, and token-specific weighting mechanism to combine the output of the window-attention block and output attention.

Detailed experimental results show ParallelTime achieves SOTA performance on real-world benchmarks.

**Strengths:**

1. This paper is the first to combine the mamba and transformer architectures for time-series, expecting short- and long-term dependencies gains from their architectures.
1. A novel Parallel Weighter is proposed to combine the outputs from the mamba blocks and the local attention blocks, and experiment results show that it benefits the performance by simply summing or averaging (with ablation studies for justification).
1. Detailed experiments are conducted to show that the Parallel Time model achieves state-of-the-art performance, with nearly half the FLOPs compared to PatchTST.
1. The writing is detailed and easy to follow.

**Weaknesses:**

This paper is solid and well-written, except for some minor weaknesses:
1. Missing definition of $\mathbf{y}_t$ in line 222, and missing relationship between $\text{Mamba}(\cdot)$ and $\mathbf{x}, \mathbf{y}, \mathbf{h}$ in line 227.
1. Missing comparison of pure mamba and pure window-attention in Figure 6.
1. Efficiency comparison only compares with PatchTST, missing some potentially more lightweight models like DLinear.
1. ParallelTime was designed to capture more long-time dependencies (or global dependencies), but the result in Table 1 shows it only performs the best half of the dataset in prediction with a prediction length of 720.
1. Figure 4 shows the visualization of the weighting outputs of the Parallel Weighter to justify that the Parallel Weighter can dynamically adjust weights of short- and long-term dependences. However, the prediction length of 192 is not long enough (compared to 720) to show long-term dependencies.

**Questions:**

1. Is there any prior work that applies window-attention with global registers in time-series data (I did not see one in the related works from line 108 to line 113)? If so, has the paper compared such a model? If not, will it work well without a combination of mamba blocks? Also in Appendix 9.3, it's stated that global registers yield slight performance enhancements. Are there any experimental results comparing ParallelTime with and without global registers?
1. What is the reason to compress the dimensionality from $\text{dim}$ to $\sqrt{\text{dim}}$ in ParallelTime Weighter other than some other dimensionalities?
1. Efficiency comparison shows that ParallelTime has fewer FLOPs (and parameters), but how about the inference latency?
1. ParallelTime Weighter outperforms simple sum or average, but how about concat (in dimension)?
1. Since window-attention has a global register, can it also capture long-term dependencies (see statements in lines 247, the author says window-attention with global registers only captures short-term dependencies).

---

> ### Author Response · Authors · 2025-11-21
> **point-by-point reply part 1**
>
> We extend our sincere gratitude to Reviewer vQqx for their thorough review and perceptive feedback. We have incorporated these suggestions into a revised version of the manuscript.
>
> W1 (Missing definition of y_t): We will add this, thank you!
>
> W2 (missing compression of pure mamba and pure attention): In our paper, we show how our method is much better than one of the best mamba-based time series models:  S-Mamba [1] and one of the best attention-based models, PatchTST [2], in Table 1.
>
> In addition, we will include the following table, in which we retrain the model by replacing the ParallelTime block with attention-only and Mamba-only variants. The results are shown below. Our model yields better results in all datasets where a large number of timestamps is used for training (electricity, traffic, weather). In the other dataset (ETT), where the data points are small, the results are mixed. As we state in our conclusion, we believe that the Parallel Weigher can be very successful as a foundation for time series models. Thank you for bringing this up. We will include this table in our paper!
>
>
> |                      |   ParallelTime |   ParallelTime |   attention |   attention |   mamba |   mamba |
> |:---------------------|--------------------------:|--------------------------:|-----------------------:|-----------------------:|-------------------:|-------------------:|
> |                      |   MSE |   MAE |   MSE |   MAE |   MSE |   MAE |
> | ('electricity', 96)  | **0.127928** | **0.222099** | 0.128914 | 0.222445 | 0.129143 | 0.224978 |
> | ('electricity', 192) | 0.147846 | **0.241042** | 0.148369 | 0.241108 | **0.146882** | 0.242145 |
> | ('electricity', 336) | 0.163897 | **0.258374** | 0.166179 | 0.259776 | **0.163415** | 0.259596 |
> | ('electricity', 720) | **0.197931** | **0.288972** | 0.19959 | 0.290493 | 0.199651 | 0.291929 |
> | ('traffic', 96)      | **0.349841** | **0.231143** | 0.353563 | 0.233127 | 0.364035 | 0.244112 |
> | ('traffic', 192)     | **0.371046** | **0.240327** | 0.373405 | 0.243209 | 0.383265 | 0.251615 |
> | ('traffic', 336)     | 0.388667 | **0.250633** | **0.388387** | 0.253613 | 0.391563 | 0.256112 |
> | ('traffic', 720)     | **0.429942** | **0.274369** | 0.43038 | 0.276113 | 0.436759 | 0.279574 |
> | ('weather', 96)      | **0.145717** | **0.189662** | 0.146944 | 0.191586 | 0.149348 | 0.194281 |
> | ('weather', 192)     | **0.189806** | **0.232382** | 0.191688 | 0.234239 | 0.193433 | 0.235974 |
> | ('weather', 336)     | **0.242879** | **0.273035** | 0.244526 | 0.275252 | 0.243764 | 0.274535 |
> | ('weather', 720)     | 0.323511 | 0.331001 | **0.317896** | **0.32952** | 0.320841 | 0.330499 |
> | ('ETTh1', 96)        | 0.365645 | 0.398115 | 0.368454 | 0.398398 | **0.363999** | **0.395025** |
> | ('ETTh1', 192)       | 0.39935 | 0.415497 | 0.413961 | 0.427931 | **0.399064** | **0.413634** |
> | ('ETTh1', 336)       | **0.385301** | **0.414899** | 0.407235 | 0.43215 | 0.38875 | 0.418704 |
> | ('ETTh1', 720)       | **0.420452** | **0.443927** | 0.454514 | 0.469012 | 0.451458 | 0.46429 |
> | ('ETTh2', 96)        | **0.262745** | 0.328645 | 0.26351 | **0.328278** | 0.264874 | 0.329455 |
> | ('ETTh2', 192)       | 0.322967 | 0.368358 | 0.324503 | 0.368115 | **0.322532** | **0.367442** |
> | ('ETTh2', 336)       | **0.312345** | **0.370723** | 0.314258 | 0.371006 | 0.318166 | 0.373097 |
> | ('ETTh2', 720)       | 0.399933 | 0.434072 | **0.397953** | **0.43328** | 0.405937 | 0.43807 |
> | ('ETTm1', 96)        | **0.28487** | **0.337948** | 0.286231 | 0.341464 | 0.29106 | 0.33942 |
> | ('ETTm1', 192)       | 0.329177 | 0.366065 | **0.327874** | 0.366504 | 0.329152 | **0.363234** |
> | ('ETTm1', 336)       | 0.365161 | 0.391733 | **0.354601** | 0.384871 | 0.359449 | **0.383618** |
> | ('ETTm1', 720)       | **0.424969** | 0.430906 | 0.425285 | 0.427482 | 0.430536 | **0.42393** |
> | ('ETTm2', 96)        | **0.162267** | **0.252789** | 0.163287 | 0.25284 | 0.163105 | 0.253653 |
> | ('ETTm2', 192)       | 0.21899 | 0.291039 | 0.219202 | **0.290193** | **0.216836** | 0.290547 |
> | ('ETTm2', 336)       | 0.276228 | 0.327567 | 0.273177 | 0.324154 | **0.270519** | **0.323236** |
> | ('ETTm2', 720)       | 0.35681 | 0.380052 | 0.355661 | 0.37955 | **0.350588** | **0.374567** |

---

> ### Author Response · Authors · 2025-11-21
> **point-by-point reply part 2**
>
> W3 (didn't check efficiency against DLinear): DLinear is a linear predictor, which makes it faster than ParallelTime, but its performance is far worse. We demonstrate that our model is highly accurate and outperforms SOTA models, such as PatchTST.
>
> | Dataset   | Pred Len | ParallelTime MSE          | ParallelTime MAE          | DLinear MSE | DLinear MAE |
> |-----------|----------|---------------------------|---------------------------|-------------|-------------|
> | **Weather** | 96     | **0.145**                | **0.189**                | 0.176      | 0.237      |
> |           | 192    | **0.189**                | **0.232**                | 0.220      | 0.282      |
> |           | 336    | **0.242**                | **0.273**                | 0.265      | 0.319      |
> |           | 720    | 0.323                    | **0.331**                | 0.333      | 0.362      |
> | **ETTh1** | 96     | **0.365**                | 0.398             | 0.375      | 0.399      |
> |           | 192    | **0.399**                | **0.415**                | 0.405| 0.416|
> |           | 336    | **0.385**                | **0.414**                | 0.439      | 0.443      |
> |           | 720    | **0.420**                | **0.443**                | 0.472      | 0.490      |
> | **ETTh2** | 96     | **0.262**                | **0.328**                | 0.289      | 0.353      |
> |           | 192    | **0.322**                | **0.368**                | 0.383      | 0.418      |
> |           | 336    | **0.312**                | **0.370**                | 0.448      | 0.465      |
> |           | 720    | 0.399             | 0.434             | 0.605      | 0.551      |
> | **ETTm1** | 96     | **0.284**                | **0.337**                | 0.299      | 0.343      |
> |           | 192    | 0.329             | 0.366             | 0.335      | **0.365**  |
> |           | 336    | 0.365             | 0.391                    | 0.369      | 0.386|
> |           | 720    | 0.424                    | 0.430                    | 0.425      | 0.421      |
> | **ETTm2** | 96     | **0.162**                | **0.252**                | 0.167      | 0.269      |
> |           | 192    | **0.218**                | **0.291**                | 0.224      | 0.303      |
> |           | 336    | 0.276                    | **0.327**                | 0.281      | 0.342      |
> |           | 720    | **0.356**                | **0.380**                | 0.397      | 0.421      |
> | **Illness** | 24   | **1.166**                | **0.657**                | 2.215      | 1.081      |
> |           | 36     | **1.293**                | **0.727**                | 1.963      | 0.963      |
> |           | 48     | **1.399**                | **0.772**                | 2.130      | 1.024      |
> |           | 60     | 1.615             | 0.844             | 2.368      | 1.096      |
> | **Electricity** | 96 | **0.128**                | **0.222**                | 0.140      | 0.237      |
> |           | 192    | **0.148**                | **0.240**                | 0.153| 0.249      |
> |           | 336    | **0.163**                | **0.258**                | 0.169      | 0.267      |
> |           | 720    | **0.197**                | **0.288**                | 0.203      | 0.301      |
> | **Traffic** | 96   | **0.349**                | **0.231**                | 0.410      | 0.282      |
> |           | 192    | **0.371**                | **0.240**                | 0.423      | 0.287      |
> |           | 336    | **0.388**                | **0.250**                | 0.436      | 0.296      |
> |           | 720    | **0.429**                | **0.274**                | 0.466      | 0.315      |
>
>
> W4 (ParalleTime in long-term forecasting): When predicting future values, the term "long" can be confusing for non-stationary data; all prediction lengths (96, 192, 366, 720) are considered long-term, even for cyclical data.
>
> W5 (need to add Parallel Weigher graph for longer prediction length (720)): We will add this, thank you for the great feedback!
>
> [1] Is Mamba Effective for Time Series Forecasting?
> [2] A Time Series is Worth 64 Words: Long-term Forecasting with Transformers

---

> > ### Author Response · Authors · 2025-11-21
> > **point-by-point reply - questions**
> >
> > Questions
> >
> > Q1 (Is window attention with registers work without Mamba? Is there results without the registers?):
> > |                      |   ParallelTime |   ParallelTime |   ParallelTime_no_reg |   ParallelTime_no_reg |   attention_with_reg |   attention_with_reg |
> > |:---------------------|--------------------------:|--------------------------:|-----------------------:|-----------------------:|-------------------:|-------------------:|
> > |                      |   MSE |   MAE |   MSE |   MAE |   MSE |   MAE |
> > | ('electricity', 96)  | **0.127928** | 0.222099 | 0.128254 | **0.222022** | 0.12893 | 0.222384 |
> > | ('electricity', 192) | **0.147846** | 0.241042 | 0.148698 | **0.240909** | 0.149374 | 0.241737 |
> > | ('electricity', 336) | **0.163897** | **0.258374** | 0.164452 | 0.25853 | 0.166482 | 0.259647 |
> > | ('electricity', 720) | 0.197931 | 0.288972 | **0.1958** | **0.28777** | 0.200678 | 0.291091 |
> > | ('traffic', 96)      | 0.349841 | 0.231143 | **0.349444** | **0.230726** | 0.354135 | 0.23399 |
> > | ('traffic', 192)     | **0.371046** | **0.240327** | 0.371078 | 0.24116 | 0.374447 | 0.243568 |
> > | ('traffic', 336)     | 0.388667 | **0.250633** | 0.38973 | 0.252111 | **0.388314** | 0.25314 |
> > | ('traffic', 720)     | **0.429942** | **0.274369** | 0.430469 | 0.274832 | 0.430781 | 0.27567 |
> > | ('weather', 96)      | **0.145717** | **0.189662** | 0.145923 | 0.190524 | 0.145848 | 0.190352 |
> > | ('weather', 192)     | **0.189806** | **0.232382** | 0.191951 | 0.233995 | 0.190269 | 0.232842 |
> > | ('weather', 336)     | 0.242879 | **0.273035** | 0.244015 | 0.275675 | **0.242812** | 0.274744 |
> > | ('weather', 720)     | 0.323511 | 0.331001 | **0.322331** | **0.330474** | 0.323941 | 0.334234 |
> > | ('ETTh1', 96)        | 0.365645 | 0.398115 | **0.364264** | **0.397478** | 0.3673 | 0.398513 |
> > | ('ETTh1', 192)       | **0.39935** | **0.415497** | 0.40521 | 0.421061 | 0.400317 | 0.416488 |
> > | ('ETTh1', 336)       | **0.385301** | **0.414899** | 0.399703 | 0.426818 | 0.410032 | 0.433871 |
> > | ('ETTh1', 720)       | **0.420452** | **0.443927** | 0.437497 | 0.457225 | 0.456617 | 0.469534 |
> > | ('ETTh2', 96)        | 0.262745 | 0.328645 | **0.260408** | **0.325749** | 0.262554 | 0.327874 |
> > | ('ETTh2', 192)       | 0.322967 | 0.368358 | **0.318956** | **0.364202** | 0.321439 | 0.366019 |
> > | ('ETTh2', 336)       | **0.312345** | 0.370723 | 0.313402 | 0.370451 | 0.313816 | **0.370424** |
> > | ('ETTh2', 720)       | **0.399933** | **0.434072** | 0.409034 | 0.441729 | 0.40163 | 0.435065 |
> > | ('ETTm1', 96)        | **0.28487** | **0.337948** | 0.288122 | 0.341916 | 0.289492 | 0.341942 |
> > | ('ETTm1', 192)       | 0.329177 | 0.366065 | 0.330998 | 0.369284 | **0.326556** | **0.365104** |
> > | ('ETTm1', 336)       | 0.365161 | 0.391733 | 0.357275 | 0.38693 | **0.356295** | **0.384818** |
> > | ('ETTm1', 720)       | 0.424969 | 0.430906 | 0.436661 | 0.435038 | **0.420379** | **0.42526** |
> > | ('ETTm2', 96)        | **0.162267** | 0.252789 | 0.162634 | **0.252642** | 0.163026 | 0.253236 |
> > | ('ETTm2', 192)       | 0.21899 | 0.291039 | 0.219589 | 0.291408 | **0.218081** | **0.289626** |
> > | ('ETTm2', 336)       | 0.276228 | 0.327567 | 0.277604 | 0.328394 | **0.273163** | **0.324289** |
> > | ('ETTm2', 720)       | 0.35681 | **0.380052** | **0.356438** | 0.382286 | 0.356559 | 0.380273 |
> >
> >
> > Q2 (Why we chose the dimension of ParallelWeigther to be $\sqrt{dim}$): We did not cross-validate it; we simply followed our intuition to use a much smaller dimension in order to compress the data. This intuition is inspired by the Squeeze-and-Excitation paper [1] where they compress the data to a much smaller dimension.
> >
> > Q3 (ParallelTime does much less FLOPs, what about latency?): In all the models that we have been compared to, including Parallel Time, the inference and the training go through the same code, hence the latency of the inference and the training is the same for all models.
> >
> > Q4 (Why not to contact the dimensions of attention and mamba components): Our model dimension is constant. If we were to concatenate the dimensions, each layer's dimension would be twice that of the previous layer, which isn't scalable and would dramatically increase the model's parameter count. Our Parallel Weighter prevents it while increasing accuracy.
> >
> >
> > Q5 (Is the window attention capture long-term dependencies due to registers?): Good point, but the registers do not leverage long-term dependencies; instead, they use global short-term dependencies, which are derived from all the short-term dependencies (within the small window) captured across the model's entire training data.
> >
> > [1] Squeeze-and-Excitation Networks

---

> ### Author Response · Authors · 2025-11-27
>
> Dear Reviewer vQqx,
>
> Thank you again for your thorough and highly insightful review. Your questions and comments have been extremely valuable and have already substantially improved the quality of our paper.
> We hope that our detailed responses below fully address all the technical concerns you raised.
>
> Should any point remain unclear, or if you believe additional experiments or clarifications would be helpful before finalizing your score, please do let us know, we would be happy to provide them promptly.
>
>
> Thank you once again for your time and expertise!
>
> Best regards,
> The Authors

---

### Official Review · Reviewer_hTqK · 2025-10-31

**Soundness:** 3
**Presentation:** 3
**Contribution:** 3
**Rating:** 6
**Confidence:** 3

**Summary:**

In this paper, the authors proposed a new parallel hybrid framework, ParallelTime, which dynamically balances short- and long-term temporal dependencies for long sequence time-series forecasting. The model integrates a Mamba branch for capturing long-term dependencies and a windowed attention branch for modeling short-term patterns, and introduces a ParallelTime Weighter that adaptively assigns token-level weights to fuse the two representations. Additionally, the authors designed an efficient Expand–Compress–Project projection strategy to reduce parameters and computational cost. Extensive experiments on multiple benchmark datasets demonstrate that ParallelTime achieves superior forecasting accuracy and efficiency compared to state-of-the-art methods.

**Strengths:**

1. The paper as a whole is content-rich, with strong logical connections between the parts. The entire work forms a closed-loop logic of problem, solution, verification, and explanation. The explanation section, in particular, discusses the underlying logic of the patterns and method performance, which enhances the interpretability of the approach.

2. The paper demonstrates good originality, both in the global design of the ParallelTime framework and in specific details (for example, Section 3.1 designs non-overlapping data blocks to ensure performance, and Section 3.4 ECP is not a simple dimensionality reduction, but uses local convolution to preserve temporal bias, etc.).

3. In Section 5 and the Appendix, the input length and patch length in the comparative experiments are explicitly aligned, eliminating the pseudo-advantage of “more input or longer context leading to seemingly better performance.” This design is well-executed and makes the performance improvement more credible.

**Weaknesses:**

1. In Section 3.3, when introducing global registers, the paper defines the content stored as global shared information, but does not indicate the rule or type of this information selection, such as whether it is statistical features like mean values or periodic values like peaks, making the register a black-box component.

2. The paper only verifies the efficiency of the method from the perspective of experimental FLOPs, lacking a theoretical explanation of the method’s superiority. For example, it could compare the time complexity from a theoretical perspective with that of transformers or other models, which would make the argument more solid.

3. The interaction process between components, such as their collaborative mechanism, is not clearly explained. For example, in Section 5.2, it is found that the Layer-2 attention weight is always higher than Layer-1, but the reason for this is not explained. Is it because Layer-1’s long-term dependency extraction provides the foundation for Layer-2’s short-term optimization, or is it because Layer-2’s FFN enhances the attention features?

**Questions:**

1. In Section 3.3, a fixed window size (a ratio of 1:9 relative to the number of input patches) is adopted, but the motivation for choosing this ratio is not explained. Was this ratio determined through experimental tuning? Is it stable across different datasets?

2. In Figure 8, the weight heatmap shows that the model assigns higher weights at mutation points. Does this pattern also appear in other datasets, such as Traffic or Exchange? Have you observed any failure cases or abnormal weight allocations?

3. The model shows the most significant improvement in long-horizon tasks. Is there direct evidence proving that the improvement comes mainly from Mamba’s long-term modeling contribution, or from the Weighter’s dynamic fusion? Is there an ablation table or figure that can verify this point?

---

> ### Author Response · Authors · 2025-11-19
> **point-by-point reply part 1**
>
> We would like to sincerely thank Reviewer hTqK for providing a detailed review and insightful comments. We have revised our paper accordingly.
>
> **Questions**
>
> Q1 (How the window size is chosen, is it stable?): We chose a small window size, which we did not cross-validate. Our goal was to keep the attention component small, allowing Mamba to summarize the long-range dependencies.
> To assess stability, we have developed an experiment that will be included in the paper. For three different window sizes: 32, 64 (which was the default in our model), and 128. it is easy to see that the window size hyperparameter is very stable, we are going to add this test to our paper. Thank you!
> |          Window Size            |   32   |     64|   128 |    32 |     64 |     128 |
> |:---------------------|-------------:|-------------:|-------------:|-------------:|-------------:|-------------:|
> |                      |  -  |    defualt |  - |   - |    defualt |  -   |
> |      (dataset, pred_len)                |   **MSE**    |      **MSE** |     **MSE** | **MAE** | **MAE** |   **MAE** |
> | ('ETTh1', 96)        | 0.398443 | 0.398115 | 0.397403 | 0.365903 | 0.365645 | 0.365505 |
> | ('ETTh1', 192)       | 0.415522 | 0.415497 | 0.416777 | 0.399645 | 0.39935 | 0.399826 |
> | ('ETTh1', 336)       | 0.414596 | 0.414899 | 0.417629 | 0.384878 | 0.385301 | 0.388219 |
> | ('ETTh1', 720)       | 0.444451 | 0.443927 | 0.447108 | 0.421574 | 0.420452 | 0.424169 |
> | ('ETTh2', 96)        | 0.328849 | 0.328645 | 0.329874 | 0.263339 | 0.262745 | 0.265131 |
> | ('ETTh2', 192)       | 0.367753 | 0.368358 | 0.367281 | 0.322318 | 0.322967 | 0.321159 |
> | ('ETTh2', 336)       | 0.371104 | 0.370723 | 0.372056 | 0.313102 | 0.312345 | 0.313522 |
> | ('ETTh2', 720)       | 0.434748 | 0.434072 | 0.435013 | 0.401034 | 0.399933 | 0.40046 |
> | ('ETTm1', 96)        | 0.339078 | 0.337948 | 0.338553 | 0.285737 | 0.28487 | 0.287112 |
> | ('ETTm1', 192)       | 0.36588 | 0.366065 | 0.367591 | 0.327836 | 0.329177 | 0.329921 |
> | ('ETTm1', 336)       | 0.391333 | 0.391733 | 0.389704 | 0.364711 | 0.365161 | 0.363135 |
> | ('ETTm1', 720)       | 0.432252 | 0.430906 | 0.42682 | 0.428861 | 0.424969 | 0.422341 |
> | ('ETTm2', 96)        | 0.252913 | 0.252789 | 0.252831 | 0.162602 | 0.162267 | 0.162515 |
> | ('ETTm2', 192)       | 0.290624 | 0.291039 | 0.291171 | 0.218898 | 0.21899 | 0.220788 |
> | ('ETTm2', 336)       | 0.327269 | 0.327567 | 0.328627 | 0.275984 | 0.276228 | 0.276237 |
> | ('ETTm2', 720)       | 0.381243 | 0.380052 | 0.380544 | 0.358742 | 0.35681 | 0.361099 |
> | ('electricity', 96)  | 0.222187 | 0.222099 | 0.222400 | 0.128244 | 0.127928 | 0.128237 |
> | ('electricity', 192) | 0.240712 | 0.241042 | 0.240385 | 0.148203 | 0.147846 | 0.147528 |
> | ('electricity', 336) | 0.259333 | 0.258374 | 0.259135 | 0.16548 | 0.163897 | 0.164615 |
> | ('electricity', 720) | 0.289634 | 0.288972 | 0.288445 | 0.198188 | 0.197931 | 0.197008 |
> | ('traffic', 96)      | 0.230634 | 0.231143 | 0.231271 | 0.350711 | 0.349841 | 0.35085 |
> | ('traffic', 192)     | 0.240041 | 0.240327 | 0.239892 | 0.37032 | 0.371046 | 0.370342 |
> | ('traffic', 336)     | 0.251109 | 0.250633 | 0.251297 | 0.390656 | 0.388667 | 0.390289 |
> | ('traffic', 720)     | 0.273737 | 0.274369 | 0.274493 | 0.43025 | 0.429942 | 0.430037 |
> | ('weather', 96)      | 0.188863 | 0.189662 | 0.189244 | 0.144705 | 0.145717 | 0.14521 |
> | ('weather', 192)     | 0.231826 | 0.232382 | 0.233043 | 0.189602 | 0.189806 | 0.190845 |
> | ('weather', 336)     | 0.273532 | 0.273035 | 0.276425 | 0.243114 | 0.242879 | 0.24714 |
> | ('weather', 720)     | 0.331901 | 0.331001 | 0.333136 | 0.323171 | 0.323511 | 0.328908 |
>
> Q2 (Higher weight for long-term dependencies at mutation points): across all the datasets, the preceding token to the mutation point gets a higher mamba component (longer dependencies are preferred), which shows how our model tries not only to look at the mutation, but what happened in the past, which makes sense.

---

> ### Author Response · Authors · 2025-11-19
> **point-by-point reply part 2**
>
> Q3 (Is the Dynamic Weigher really helpful for performance?): In our paper, we show how our method is much better than one of the best mamba-based time series models:  S-Mamba [1] and one of the best attention-based models, PatchTST [2], in Table 1.
>
> In addition, we will include the following table, in which we retrain the model by replacing the ParallelTime block with attention-only and Mamba-only variants. The results are shown below. Our model yields better results in all datasets where a large number of timestamps is used for training (electricity, traffic, weather). In the other dataset (ETT), where the data points are small, the results are mixed. As we state in our conclusion, we believe that the Parallel Weigher can be very successful as a foundation for time series models.
>
>
> |                      |   ParallelTime |   ParallelTime |   attention |   attention |   mamba |   mamba |
> |:---------------------|--------------------------:|--------------------------:|-----------------------:|-----------------------:|-------------------:|-------------------:|
> |                      |   MSE |   MAE |   MSE |   MAE |   MSE |   MAE |
> | ('electricity', 96)  | **0.127928** | **0.222099** | 0.128914 | 0.222445 | 0.129143 | 0.224978 |
> | ('electricity', 192) | 0.147846 | **0.241042** | 0.148369 | 0.241108 | **0.146882** | 0.242145 |
> | ('electricity', 336) | 0.163897 | **0.258374** | 0.166179 | 0.259776 | **0.163415** | 0.259596 |
> | ('electricity', 720) | **0.197931** | **0.288972** | 0.19959 | 0.290493 | 0.199651 | 0.291929 |
> | ('traffic', 96)      | **0.349841** | **0.231143** | 0.353563 | 0.233127 | 0.364035 | 0.244112 |
> | ('traffic', 192)     | **0.371046** | **0.240327** | 0.373405 | 0.243209 | 0.383265 | 0.251615 |
> | ('traffic', 336)     | 0.388667 | **0.250633** | **0.388387** | 0.253613 | 0.391563 | 0.256112 |
> | ('traffic', 720)     | **0.429942** | **0.274369** | 0.43038 | 0.276113 | 0.436759 | 0.279574 |
> | ('weather', 96)      | **0.145717** | **0.189662** | 0.146944 | 0.191586 | 0.149348 | 0.194281 |
> | ('weather', 192)     | **0.189806** | **0.232382** | 0.191688 | 0.234239 | 0.193433 | 0.235974 |
> | ('weather', 336)     | **0.242879** | **0.273035** | 0.244526 | 0.275252 | 0.243764 | 0.274535 |
> | ('weather', 720)     | 0.323511 | 0.331001 | **0.317896** | **0.32952** | 0.320841 | 0.330499 |
> | ('ETTh1', 96)        | 0.365645 | 0.398115 | 0.368454 | 0.398398 | **0.363999** | **0.395025** |
> | ('ETTh1', 192)       | 0.39935 | 0.415497 | 0.413961 | 0.427931 | **0.399064** | **0.413634** |
> | ('ETTh1', 336)       | **0.385301** | **0.414899** | 0.407235 | 0.43215 | 0.38875 | 0.418704 |
> | ('ETTh1', 720)       | **0.420452** | **0.443927** | 0.454514 | 0.469012 | 0.451458 | 0.46429 |
> | ('ETTh2', 96)        | **0.262745** | 0.328645 | 0.26351 | **0.328278** | 0.264874 | 0.329455 |
> | ('ETTh2', 192)       | 0.322967 | 0.368358 | 0.324503 | 0.368115 | **0.322532** | **0.367442** |
> | ('ETTh2', 336)       | **0.312345** | **0.370723** | 0.314258 | 0.371006 | 0.318166 | 0.373097 |
> | ('ETTh2', 720)       | 0.399933 | 0.434072 | **0.397953** | **0.43328** | 0.405937 | 0.43807 |
> | ('ETTm1', 96)        | **0.28487** | **0.337948** | 0.286231 | 0.341464 | 0.29106 | 0.33942 |
> | ('ETTm1', 192)       | 0.329177 | 0.366065 | **0.327874** | 0.366504 | 0.329152 | **0.363234** |
> | ('ETTm1', 336)       | 0.365161 | 0.391733 | **0.354601** | 0.384871 | 0.359449 | **0.383618** |
> | ('ETTm1', 720)       | **0.424969** | 0.430906 | 0.425285 | 0.427482 | 0.430536 | **0.42393** |
> | ('ETTm2', 96)        | **0.162267** | **0.252789** | 0.163287 | 0.25284 | 0.163105 | 0.253653 |
> | ('ETTm2', 192)       | 0.21899 | 0.291039 | 0.219202 | **0.290193** | **0.216836** | 0.290547 |
> | ('ETTm2', 336)       | 0.276228 | 0.327567 | 0.273177 | 0.324154 | **0.270519** | **0.323236** |
> | ('ETTm2', 720)       | 0.35681 | 0.380052 | 0.355661 | 0.37955 | **0.350588** | **0.374567** |
>
> **Weaknesses**
>
> W3 (Why the attention amount is higher in later layers): Thank you for bringing this up. We have now integrated this into the relevant section. Before entering the first ParallelBlock (which weights Mamba and attention), each token has not yet been combined with others and thus encapsulates only its own data. By the second layer, however, each token has been influenced by preceding tokens. As a result, in the first layer, our Weighter prefers to assign more weight to Mamba, which summarizes long-term dependencies. In later layers, this long-term data becomes embedded in nearby tokens, allowing us to access it via windowed attention, which receives more weight from the Parallel Weighter. We find this emergent behavior particularly interesting: the model begins by summarizing long-range dependencies and then utilizes short-range dependencies on this summarized representation to generate the prediction.

---

> ### Author Response · Authors · 2025-11-27
>
> Dear Reviewer hTqK,
>
>
> Thank you once again for your detailed and thoughtful review. Your feedback has been incredibly helpful and has already contributed significantly to strengthening our manuscript.
>
>
> We hope our responses below fully clarify all the concerns you raised. If any aspect still seems unclear or if you think further experiments or details would assist you in finalizing your evaluation, please don’t hesitate to let us know—we can provide them right away.
>
>
> We truly appreciate the time and effort you have invested in reviewing our work!
>
>
> Best regards,
> The Authors

---

> > ### Comment · Reviewer_hTqK · 2025-11-27
> >
> > Thank you for the detailed response, I will keep my positive score.

---

### Author Response · Authors · 2025-12-01
**Summary of changes**

We thank all four reviewers for their careful reading of our paper and for providing many valuable suggestions and insightful points, which we address in detail in our point-by-point responses below. The **reviewers appreciated our innovative method**, ParallelTime Weighter; one reviewer **upgraded their score from 4 to 6**, while the others were unable to respond before the block.

Most reviewers requested various ablation studies, with the most common query being a comparison between ParallelTime Weighter (ours) vs. Mamba vs. Attention. The results demonstrated that our method outperforms the baselines, especially as the number of data points increases (i.e., with more data points in the dataset).

Thanks to the reviewers' suggestions, we have revised our paper as follows:

- Added an ablation study comparing ParallelTime Weighter (ours) vs. Mamba vs. Attention (as requested by multiple reviewers), Table 6\.
- Added robustness analysis for the window size hyperparameter in Table 10\.
- Provided a clearer explanation of when higher weights for long-term dependencies occur at mutation points (Section 5.1).
- Added the definition of y\_t at line 224\.
- Explained the rationale for choosing the dimension of ParallelTime Weighter to be dim⁡\\sqrt{\\dim}dim​ (Section 3.4).
- Corrected grammatical and spelling errors throughout the paper.

---

### Meta-Review · Area_Chair_BMzY · 2025-12-28

**Summary:**

The paper proposes ParallelTime, a multi-scale hybrid forecasting model that uses Mamba to capture long-range dependencies and a Transformer to model short-term dynamics, combined via a new dynamic weighting mechanism.

My decision is driven primarily by reviewer **gvPa**’s concern, which I find valid and insufficiently addressed in the rebuttal: the paper provides neither theoretical justification nor convincing empirical evidence that the proposed parallel dual-branch design improves over established multi-scale or decomposition-based approaches. In particular, the manuscript does not meaningfully discuss or benchmark against common time-series decomposition techniques (e.g., seasonal–trend decomposition or frequency decomposition), which are widely used and often critical for strong long-horizon forecasting performance (e.g., approaches such as *TimeKAN*). Since the core novelty of ParallelTime appears to be the parallel dual-branch architecture, a direct comparison with representative decomposition schemes is important to substantiate the claimed contribution. I think Mamba or Transformer are just certain design choices for modeling short / long-term dependency.

In addition, I did not see strong endorsement from the other reviewers regarding the paper’s novelty or the strength of its empirical results. Taken together with the concerns above, I am leaning toward rejection.

**Reviewer Concerns:**

Common concern from Reviewers about the comparison with pure Mamba or pure window-attention has been addressed in Table 6.

Apart from the common concern, for reviewer **hTqK**, the concern around the mechanism of register is not addressed by the rebuttal. The other concerns from the reviewer on theoretical insight on time complexity and possible explanation of the collaborative mechanism are not addressed. The author addressed the concern on motivation for the window size with new experiments.

For reviewer **vQqx**, the minor concerns around presentation have been addressed. However, the concern on speed comparison with DLinear was not addressed in the rebuttal (The author only claimed that DLinear is faster than ParallelTime without time measurement). The reviewer also raised question about the performance of window-attention + global registry and I did not find the answer in the rebuttal.

For reviewer **gvPa**, the major concern on comparison with other decomposition strategies is not addressed.

For reviewer **oNcd**, the concern around the mechanism of ParallelTime weighter has been addressed. The reviewer also expressed concern on novelty and the author clarified that the major novelty is the Parallel Weighter component.

**Reviewer Scores:**

Reviewer oNcd raised the score from 4 to 6. I think the other reviewers may keep the score.

---

### Decision · Program_Chairs · 2026-01-26

Reject